

**Medium-range predictability of early summer sea ice thickness distribution in**
**the East Siberian Sea: Importance of dynamical and thermodynamic melting**
**processes**
**Takuya Nakanowatari[1,*], Jun Inoue[1], Kazutoshi Sato[1], Laurent Bertino[2], Jiping Xie[2], Mio**
**Matsueda[3], Akio Yamagami[3], Takeshi Sugimura[1], Hironori Yabuki[1], and Natsuhiko Otsuka[4]**
[1]National Institute of Polar Research, 10-3, Midori-cho, Tachikawa-shi, Tokyo, 190-8518, Japan;
[2]Nansen Environmental and Remote Sensing Center, Thormøhlens gate 47, N-5006 Bergen,
Norway; [3]Center for Computational Sciences, University of Tsukuba, 1-1-1 Tennodai, Tsukuba,
Ibaraki 305-8577, Japan; [4]Arctic Research Center, Hokkaido University, Kita-21 Nishi-11 Kita-ku,
Sapporo, 001-0021, Japan
*Corresponding author: Takuya Nakanowatari, E-mail: nakanowatari.takuya@nipr.ac.jp





**Abstract**
Accelerated retreat of Arctic Ocean summertime sea ice has focused attention on the potential use
of the Northern Sea Route (NSR), for which sea ice thickness (SIT) information is crucial for safe
maritime navigation. This study evaluated the medium-range (lead time below 10 days) forecast
skill of SIT distribution in the East Siberian Sea (ESS) in early summer (June–July) based on the
TOPAZ4 ice ocean data assimilation system. Comparison of the operational model SIT data to all
available observations (in situ and satellite) showed that the TOPAZ4 reanalysis reproduces the
observed seasonal cycle and the rates of advance and melting of SIT in the ESS, with average bias
of approximately ±20 cm. Pattern correlation analysis of the SIT forecast data over 4 years
(2013−2016) reveals that the early summer SIT distribution is skillfully predicted for a lead time of
up to 3 days, but that the prediction skill drops abruptly after the 4th day, which is related to
dynamical process controlled by synoptic-scale atmospheric fluctuations. For longer lead times (>4
days), the thermodynamic melting process takes over, which makes most of the remaining prediction
skill. In July 2014, during which an ice-blocking incident occurred, relatively thick SIT (~150 cm)
was simulated over the ESS, which is consistent with the reduction of vessel speed. These results
suggest that TOPAZ4 sea ice information has a great potential for practical applications in
summertime maritime navigation via the NSR.



## 1 Introduction

During recent decades, sea ice cover in the Northern Hemisphere has shown remarkable reduction and the largest rates of decrease of 100,000 km$^2$ decade$^{-1}$ has been observed in the western Arctic Ocean in summer [Cavalieri and Parkinson, 2008]. Sea ice retreat influences the light conditions for phytoplankton photosynthesis activity [Wassmann, 2011], and the resultant meltwater influences the marine environment via ocean acidification [Yamamoto-Kawai et al., 2011]. In winter, shrinkage of the sea ice area in marginal seas, such as the Barents Sea changes the surface boundary conditions of the atmosphere, influences planetary waves, and causes blocking events that are one of the possible causes of the recent severe winters in mid-latitude regions [Honda et al., 2009; Inoue et al., 2012; Mori et al., 2014; Overland et al., 2015; Petoukhov and Semenov, 2010; Screen, 2017].

In contrast to these climatic consequences and problems for the marine ecosystem caused by the reduction in sea ice, the retreat of Arctic sea ice has new opportunities for commercial maritime navigation. It has been reported that exploitation of shipping routes in the Arctic Ocean, i.e., the Northern Sea Route (NSR), could reduce the navigational distance between Europe and Asia by about 40% in comparison with routes via the Suez Canal [Schøyen and Bråthen, 2011]. Melia et al. [2016] discussed the possibility of a viable trans-Arctic shipping route in the 21st century, based on the Coupled Model Intercomparison Project Phase 5 global climate model simulation. Currently, the summertime use of the NSR by commercial vessels such as cargo ships and tankers has increased [Eguíluz et al., 2016]. Therefore, obtaining precise information on sea ice condition and evaluating the forecast skill of operational sea ice models have become urgent issues.

Many previous studies have examined the predictability of summertime sea ice change in the Arctic Ocean in terms of its coverage [Wang et al., 2013] and motion [Schweiger and Zhang, 2015]. Kimura et al. [2013] reported a good correlation of the spatial distribution of summertime sea ice concentration (SIC) with winter ice divergence/convergence. Their study indicated that sea ice thickness (SIT) or sea ice volume before the melt season is a source of predictability for summertime



SIC. Recently, their study was supported by hindcast experiments undertaken using a climate model, in which the SIC in the East Siberian Sea (ESS) was shown to have significant seasonal prediction skill [Bushuk et al., 2017]. The significant impacts of SIT condition on the seasonal prediction of SIC in the Arctic Ocean have been highlighted by many studies [Lindsay et al., 2008; Holland et al., 2011; Blanchard-Wrigglesworth and Bitz, 2014; Collow et al., 2015; Melia et al., 2015; Chen et al. 2017; Melia et al. 2017]. Thus, the persistence of SIT or sea ice volume is one of the key factors determining the skill of seasonal predictions of summertime sea ice area.

Earlier studies have focused primarily on the seasonal to interannual predictability of SIC or sea ice area in the Arctic Ocean; thus, subseasonal variation in SIT and its predictability have not been examined fully for near-term route planning. Although the summertime sea ice extent has rapidly decreased on interannual timescale, substantial sea ice area still remains in critical stretches of the NSR such as the ESS in early summer (June–July). Since precise information regarding SIT and its near-future condition is crucial for icebreaker operations [Tan et al., 2013; Pastusiak, 2016], it is important to clarify the medium-range (3 to 10 days lead time) predictability of summertime SIT in the Arctic Ocean.

Synoptic-scale fluctuations of cyclone and anticyclone is greater over the Arctic Ocean and Eurasia in summer than in winter [Serreze and Barry, 1988; Serreze and Barrett, 2008]. In recent years, there is a risk that an Arctic cyclone becomes extremely developed and covered the entire Pacific sector [Simmonds and Rudeva, 2012; Yamagami et al. 2017]. Because the ESS corresponds to the route of Arctic cyclones generated over the Eurasian Continent [Orsolini and Sorteberg, 2009], it is expected that synoptic-scale atmospheric fluctuations would influence substantially the spatial distribution of SIT and ice motion in the ESS. Ono et al. [2016] highlighted the importance of atmospheric prediction skill on medium-range forecasts of sea ice distribution in the ESS based on a case of an extreme cyclone that occurred on 6 August 2012. On the other hand, earlier studies pointed out that the sea ice melting process is important for the long-term prediction of summertime sea ice





extent [e.g., Bushuk et al., 2017]. But the relative importance of dynamical and thermodynamic
processes on the medium-range forecast skill of summertime sea ice properties has not yet been well
understood.

Since 2010, ice–ocean forecasts and a 20-years reanalysis are available for the Arctic Ocean,

based on the TOPAZ ocean data assimilation system (Towards an Operational Prediction system for
the North Atlantic European coastal Zones) in its 4th version [Sakov et al., 2012]. The Norwegian
Meteorological Institute provides 10-day forecast products in daily mean fields, forced at the surface
by the ECMWF operational atmospheric forecasts, updated daily and distributed by the Copernicus
Marine Environment Monitoring Services (http://marine.copernicus.eu). The reliability of the
corresponding TOPAZ4 reanalysis data has been evaluated previously through comparison with in
situ and satellite SIT data [Xie et al. 2017; Nakanowatari et al. 2017]. They showed the SIT in the
TOPAZ4 reanalysis data are comparable to observed values over the Beaufort Gyre and central Arctic
Ocean, although the SIT overall shows a negative bias of several dozen centimeters throughout a year.
Thus, it is expected that the SIT data in the TOPAZ reanalysis data should also be reliable in the ESS
even in the melting season, and the forecast SIT data should show skillful prediction skill on medium-
range time scale.

In this study, we examined the predictability of the early summer SIT distribution in the ESS on

the medium-range timescale and discussed its underlying physical mechanisms, based on the
TOPAZ4 forecast dataset and trivial dynamical and thermodynamical models. Section 2 describes
the data and methods. Section 3 evaluates the reliability of the SIT data in the TOPAZ4 reanalysis
data through comparison with all available in situ and satellite observations, as well as operational
model analyses, with particular emphasis on the ESS. In section 4, we examine the predictability of
the SIT distribution in the ESS based on TOPAZ4 forecast data. Section 5 examines the relationship
between sea ice conditions and vessel speed during an ice-blocking event that occurred in July 2014.
A discussion and the derived conclusions are presented in section 6.




## 2 Data and Methods


This study used daily mean sea ice data derived from the TOPAZ4 Arctic sea ice forecast system


dataset, in which the SSM/I SIC data, hydrographic temperature and salinity data, along-track sea


level anomaly, and satellite estimates of ice drift and sea surface temperature were assimilated, but


sea ice thickness was not yet assimilated in this version of the reanalysis [Simonsen et al. 2017]. The


TOPAZ4 system was designed as a regional ice–ocean coupled system forced with atmospheric flux


data. The ocean model of TOPAZ4 is based on version 2.2 of HYCOM, which uses isopycnical


vertical coordinates in the ocean interior and z level coordinates in the near-surface layer. The sea ice


model uses an elastic–viscous–plastic rheology [Hunke and Dukowicz, 1997]. The thermodynamic


processes are based on a three-layer thermodynamic model with one snow and 2 ice layers [Semtner,


1976] with a modification for subgrid-scale ice thickness heterogeneities [Fichefet and Morales


Maqueda, 1997]. The model domain covers the Arctic Ocean and the North Atlantic, and the lateral


boundaries are relaxed to monthly mean climatological data. The spatial resolution is 12−16 km with


28 hybrid layers, which constitutes eddy-permitting resolution in low- and mid-latitude regions but

not in the Arctic Ocean. It has been reported that the SIT of the TOPAZ4 reanalysis data has

substantial negative bias from 2001 to 2010 due to excessive snowfall, which has been modified after

2011 [Xie et al., 2017]. Therefore, this study used SIT data from 1 January 2011 to 31 December


2014.


The data assimilation method of TOPAZ4 is a deterministic version of the ensemble Kalman


filter (EnKF) [Sakov and Oke, 2008] with an ensemble of 100 dynamical members. Since EnKFs


have time-dependent state error covariances, this method is suitable for data assimilation of


anisotropic variables in areas close to the sea ice edge [Lisæter et al. 2003, Sakov et al. 2012]. In this


system, in situ hydrographic observations are assimilated together with satellite observations of the


ocean such as sea surface temperature and sea surface height. Since this system assimilates the SIC





and sea ice velocity (but the latter only in cold season), one should expect adequate simulation of SIT
through the ridging process [Stark et al. 2008]. The TOPAZ4 reanalysis data were produced forced
with 6-hourly atmospheric fluxes from the ERA Interim reanalysis [Dee et al., 2011]. The surface
turbulent heat flux and momentum flux were both calculated using bulk formula parameterizations
[Kara et al., 2000; Large and Pond, 1981]; thus, fluxes derived from the atmospheric model were not
used. The forecast and reanalysis systems have almost the same settings and their results are similar
during their overlap period (not shown).

To evaluate the prediction skill of the TOPAZ4 forecast system, we used daily mean sea ice

forecast data from 2012 to 2016 [Simonsen et al. 2017]. A probabilistic 10-member ensemble forecast
was performed with the ECMWF medium-range (up to 10 days) atmospheric forecast data updated
daily, out of which only the ensemble average is used. We excluded the forecast data of 2012 in this
study, because the sea ice coverage of the ESS in early summer was quite small. Since the forecast
data were only provided weekly before 2016, the total of 259 cases was assembled during the study
period. The skill core was quantified using pattern correlation coefficients (PCCs), which are used
widely in deterministic forecast verification [Barnett and Schlesinger, 1987]:
$$PCC = \frac{\sum_{ij=1}^{N}\left(f_{ij} - \overline{f}_{ij}\right)\left(a_{ij} - \overline{a}_{ij}\right)}{\sqrt{\sum_{ij=1}^{N}\left(f_{ij} - \overline{f}_{ij}\right)^2}\sqrt{\sum_{ij=1}^{N}\left(a_{ij} - \overline{a}_{ij}\right)^2}} \tag{1}$$

where $f_{ij}$ and $a_{ij}$ are forecast and analysis sea ice variables, respectively. The overbar denotes the
average values over the analyzed area (see Fig. 1a); thus the PCC reflects the correlation of observed
and signal anomalies relative to their respective spatial means.

To evaluate the reliability of the SIT values in the TOPAZ4 reanalysis data during the freezing

season, we mainly used the merged product of CryoSat-2 (CS2) and the Soil Moisture and Ocean
Salinity (SMOS) SIT products (hereafter, CS2SMOS) from 2011 to 2014 [Ricker et al. 2017], which
were provided by the online sea-ice data platform "meereisportal.de" [Grosfeld et al. 2016]. These



data are interpolated to 25-km resolution based on optimal interpolation and they are available from
October to April. In general, CS2 data have large uncertainty in the estimation of SIT of <1 m, while
the SMOS relative uncertainties are lowest for very thin ice. Thus, the merged product is – to date –
considered the best estimate of the SIT distribution across the entire Arctic Ocean, including the ESS.
For the melting season (May–July), there is no reliable estimate of SIT distribution in the ESS,
we therefore used only in situ SIT data of autonomous ice mass balance (IMB) buoys obtained
between 26 March and 29 July 2014 near the ESS [Perovich et al., 2013]. To compare the two-
dimensional SIT data with IMB buoy data, we re-gridded the gridded SIT data along the IMB buoy
trajectories. This comparison method is almost identical to that adopted by Sato and Inoue [2017]
who compared IMB buoy data with SIT data of the NCEP-CFSR reanalysis. As a reference for SIC,
we used daily mean SIC data derived from AMSR2 passive microwave radiometer sensors using the
bootstrap algorithm [Comiso and Nishio, 2008; JAXA, 2013].
As an alternative model reanalysis, we used the PIOMAS outputs, which are derived from the
coupled ice–ocean modeling and assimilation system based on the Parallel Ocean Program POP and
the Thickness and Enthalpy Distribution (TED) sea ice model, forced with NCEP-NCAR reanalysis
data [Zhang et al., 2003]. In this dataset, SIC and sea surface temperature are assimilated by adoptive
nudging, and many studies [Schweiger et al., 2011; Lindsay and Zhang, 2006; Stroeve et al., 2014]
have compared PIOMAS output with observed SIT data and found it the most reliable estimate of
observed SIT in the Arctic Ocean [Laxon et al., 2013; Wang et al. 2016]. The temporal and horizontal
resolutions of the observed and simulated SIT data are summarized in Table 1. Before comparing the
gridded SIT data with IMB buoy data in each grid point, we reconstructed these SIT data on a 0.25°
latitude–longitude grid by applying bilinear interpolation.
To examine the source of medium-range predictability in SIT distribution, we also used
ECMWF atmospheric forecast data on a 1.25° latitude–longitude grid from 2013 to 2016, derived
from the THORPEX Interactive Grand Global Ensemble through its data portal




(http://tigge.ecmwf.int). This dataset is very similar to the atmospheric forecast data used for the
TOPAZ4 operational forecast system [Simonsen et al. 2017]. For the examination of atmospheric
forecast skill, we used 51 ensemble daily means of zonal and meridional wind speed at 10-m height
on the same days for the TOPAZ4 forecast data at lead times of 0−10 day.
To evaluate the influence of sea ice condition on vessel speed in the ESS, we used Automatic
Identification System (AIS) data from two tankers during their passage through the ESS on 4−26 July
2014, which were provided by Shipfinder (http://jp.shipfinder.com/). Their ice classes correspond to
IA Super in the Finnish–Swedish Ice Class Rules, and these vessels are capable of navigating sea ice
regions in which SIT is up to 50–90 cm. Both tankers were likely to be hindered considerably by ice
conditions, even under escort by Russian nuclear-powered ice-breakers; thus, these AIS data are
considered suitable for a case study of the influence of SIT on icebreaker speed.

**3 Comparisons between TOPAZ4 and other available SIT data**
Figure 1a shows the spatial distribution of observed (CS2SMOS) SIT in April (when SIT is
maximum) in the Arctic marginal seas of the Laptev Sea, ESS, and Chukchi sea. The sea ice
observations show the maximum thickness (>3 m) near Greenland, but relatively thick ice (~1.8 m)
can also be found around the ESS. These features are qualitatively simulated in the TOPAZ4
reanalysis data (Fig. 1b). The differences in SIT between the TOPAZ4 reanalysis and CS2SMOS data
reveal remarkable negative bias (i.e., smaller than −0.8 m) in the TOPAZ4 reanalysis in the central
Arctic Ocean (Fig. 1c); however, the magnitude of the negative bias is smaller in coastal areas such
as the ESS. The PCC of SIT between TOPAZ4 and CS2SMOS in the Arctic marginal seas (65°−80°N,
80°E−160°W, shown in Fig. 1a) is 0.89 in April, which is comparable with that between the PIOMAS
output and CS2SMOS (Table 2). The PCCs in other months are also comparable with those of the
PIOMAS output. It should be noted that a larger positive bias in TOPAZ4 is located solely in the
region of the Beaufort Gyre, with about 50 cm excess thickness (Fig. 1c). This positive bias is



however consistent with the large underestimation of CS2SMOS SIT over the Beaufort Sea, which is
related to the existence of heavily deformed ice [Ricker et al. 2017].

Figure 2 shows the time series of daily mean SIT derived from CS2SMOS, TOPAZ4 reanalysis,

and PIOMAS output, averaged over the ESS (70°–80° N, 150°–180° E, shown in Fig. 1a). The
TOPAZ4 SIT data are reasonably similar to the seasonal cycle of CS2SMOS data with maxima in
April–May and minima in October–November, although the TOPAZ4 SIT data at the beginning of
2011 are highly underestimated. This might be related to the persistence of the negative bias until
2010 [Xie et al., 2017]. The monthly mean biases of TOPAZ4 SIT data relative to CS2SMOS are less
than −23 cm in March and April (Table 3). Thus, even though the negative bias in the TOPAZ4
reanalysis data is relatively large in the central Arctic (Fig. 1c), the TOPAZ4 SIT is comparable with,
or larger than, the CS2SMOS data over the Arctic marginal seas. The PIOMAS SIT also follows the
seasonal cycle of CS2SMOS data, but it is overestimated somewhat from January to May (Fig. 2).
The mean biases of PIOMAS SIT relative to CS2SMOS are 48 and 66 cm in March and April,
respectively, which are much larger than for TOPAZ4 (Table 3). In the melting season (May–July),
the seasonal reduction in SIT of TOPAZ4 near the ESS is comparable with that observed in the IMB
buoy data (Fig. 3). The TOPAZ4 SIT has weak positive bias of <25 cm relative to the IMB buoy data
from May to July (Table 3), which is smaller than for PIOMAS. Consequently, the TOPAZ4 SIT in
the ESS can be considered successful in simulating the seasonal cycles of CS2SMOS and IMB buoy
data within the range of approximately ±20 cm, which is lower than the negative bias found in the
central Arctic Ocean. The errors in the central Arctic Ocean and Beaufort Sea are probably larger
because they contain older multi-year ice for which the SIT errors have accumulated errors in sea ice
drift and thermodynamics over longer times.



### 4. Medium-range forecast skill of SIT distribution in the ESS


In this section, we evaluate the prediction skill of SIT based on the PCCs between the analysis
and predicted data in the ESS. However, before this evaluation, we examine the mean fields and the
variability of the SIT and SIC distributions in early summer. Figure 4a presents the spatial distribution
of the climatological SIT and SIC in July, which shows that relatively thick sea ice (~1 m) covers
50%−70% of the ESS. Along the zone of the sea ice edge, the temporal standard deviation of the daily
mean SIT anomaly is relatively large with the maximum value of 0.6 m in the coastal region (Fig.
4b) and the area-averaged value is maximum in July−August (Fig. 4c). Since the SIT reduction rate
in the ESS is strongest in these months (Fig. 4c) and the storm activity is prevalent for periods of
several days [Orsolini and Sorteberg, 2009], it is likely that dynamical and thermodynamically-
induced SIT variations are large. Note that the RMS of the SIC anomaly averaged over the ESS also
shows a similar seasonal cycle (not shown). Thus, it is meaningful to examine the medium-range
predictability of early summer SIT distribution in the ESS.
Figure 5 shows the seasonal dependency of PCC between the predicted and analyzed SIT at lead
times of 0−9 days. We found that the overall prediction skill is relatively low in June-July with a
larger spread compared with the cold season (January−May), which is consistent with the larger
variance of the SIT anomaly in the ESS (Fig. 4c). In early summer (June−July), the SIT distribution
is predicted skillfully for a lead time of up to 3 days (Fig. 6); however, the prediction skill decreases
abruptly at a lead time of 4 days, in which the standard deviation is also relatively large. Since such
an abrupt reduction of the prediction skill is also found in May and October (Fig. 5), when the
influence of sea ice melt is quite small (Fig. 4c), the abrupt reduction of early summer SIT prediction
skill might be attributable to dynamical advection of sea ice.
To examine the influence of dynamical processes on the prediction skill of early summer SIT
distribution, we consider the prediction skill of sea ice velocities and surface wind velocities. The
prediction skill of sea ice velocity stays on a high level (~0.8) with small spread for a lead time of up


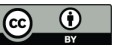

to 3 days, but decreases down to 0.6–0.7 for a lead time of 4 days (Fig. 7a). The early summer
prediction skill of surface wind speed also shows the same abrupt decrease at a lead time of 4 days,
and the rate of decrease of prediction skill is larger in meridional direction (Fig. 7b). Since the SIT
distribution has a zonally homogeneous pattern (Fig. 4a), it is suggested that the meridional
component of SIT advection is sensitive to the sea ice transport, which influences the SIT distribution
in the ESS. These results confirm that the prediction skills of the sea ice velocities are strongly related
to those of surface wind speeds in the ESS.

Figure 8 shows the temporal evolutions of SIT and ice velocity for analysis and a forecast

bulletin starting from 2nd July 2015, which is a typical case of the abrupt decrease in the prediction
skill of SIT as well as sea ice velocities for a lead time of 4 days (Fig. 8; lower panel). For lead times
of +0 (2 July) to +2 days (4 July), the spatial distributions of SIT and ice velocity are predicted
skillfully with only small differences between them (Fig. 8; right panels). At a lead time of +4 days
(6 July), the analyzed sea ice velocity is directed northwestward in the ESS, which is related to the
cyclonic circulation over the Novosibirsk Islands; however, the predicted sea ice velocity is directed
southwestward. At a lead time of +6 days, the predicted and analyzed sea ice velocity are completely
unrelated. The resultant southward anomaly of sea ice velocity leads to positive and negative
anomalies in SIT in the coastal and offshore regions, respectively. We also examined the time
evolutions of the surface wind velocities in the atmospheric forecast data, and found them very similar
to the sea ice velocity fields (not shown). These results indicate that the abrupt reduction of the
prediction skill of early summer SIT in the ESS is related to a deficiency at predicting Arctic cyclone.

Further, we examine diagnostically the ice drift speed and direction based on a classical free-

drift theory [Leppäranta, 2005], using the sea ice speed of TOPAZ4 reanalysis data and ERA interim
atmospheric wind data in July 2011− 2014. The general solution of sea ice speed ($u$) can be described
as complex numbers:





$$u = \alpha e^{-i\theta} U_a + U_{wg},$$
(2)

where $U_a$, and $U_{wg}$ are the wind speed and geostrophic water velocities, respectively. The terms $\alpha$ and
$\theta$ are the wind factor and the deviation angle of ice motion from the surface wind, respectively, where
a positive angle is in counterclockwise direction. If we neglect the geostrophic water velocity $U_{wg}$,
the wind factor and deviation angle can be obtained in the following form:
$$\alpha^4 + 2\sin\theta_w RNa\alpha^3 + R^2 Na^2\alpha^2 - Na^4 = 0,$$
(3)

$$\theta = \arctan\left(\tan\theta_w + \frac{RNa}{\alpha\cos\theta_w}\right) - \theta_a,$$
(4)

where $\theta_w$ and $\theta_a$ are the boundary layer turning angles of water and air, respectively. The turning angle
$\theta$ is the angle between the vectors of the ice–water stress and the sea ice motion, which is a
consequence of the viscous effect within the ocean boundary layer. The Nansen number $Na$ is defined
by $\sqrt{\rho_a C_a / \rho_w C_w}$, where $\rho_a$ and $\rho_w$ represent the density of air and water, respectively, and $C_a$ and
$C_w$ are air and water drag coefficients, respectively. The Rossby number $R$ is defined by
$(\rho h_{ice} f)/(\rho_w C_w Na|U_a|)$, where $\rho$ is the ice density, $f$ is the Coriolis parameter, and $|U_a|$ is the speed
of the surface wind. To calculate the wind factor $\alpha$ and the deviation angle $\theta$ under a given surface
wind speed, we used constant parameters of $C_a = 1.2 \times 10^{-3}$, $C_w = 5 \times 10^{-3}$, $\rho_a = 1.3$ kg m$^{-3}$, $\rho_w =$
1026 kg m$^{-3}$, $\rho = 910$ kg m$^{-3}$, $f = 1.3 \times 10^{-4}$ s$^{-1}$, and $\theta_w = 20°$, which are values typical of the Arctic
Ocean [McPhee, 2012]. The value of $\alpha$ was calculated numerically from a 4th-order polynomial (Eq.

(3)).

On a first order approximation, the daily mean sea ice speed is linearly proportional to the surface
wind speed (10-m height) averaged over a part of the ESS (Fig. 9a). The correlation between them is
0.96, which is significant at the 99% confidence level, based on the Monte Carlo simulation [Kaplan
and Glass, 1995]. The regression coefficient of ice speed onto the 10-m wind speed is 0.022, which





is consistent with the well-known 2% relationship between the speed of ice and the surface wind
speed [Thorndike and Colony, 1982]. The number of the TOPAZ4 ice speed data within ±20% of the
theoretical value is 79 days, which accounts for 63% of the total analyzed period. Note that the
observed regression coefficient is somewhat larger than the theoretical value (0.018) averaged over
the range of surface wind speed of 2–10 m s$^{-1}$ calculated from Eq. (2). Since the classical free drift
theory [Leppäranta, 2005] neglects both the Ekman layer velocity and the ocean geostrophic velocity,
the absence of an ice-ocean boundary layer is likely to underestimate the wind-induced ice velocity
[Park and Stewart, 2016]. The deviation angle of sea ice motion in TOPAZ4 is estimated as 20°–40°
under a wind condition >5 cm s$^{-1}$, but it gradually increases to 40°–70° under weaker wind conditions
of <5 cm s$^{-1}$ (Fig. 9b). The decrease of the deviation angle as the surface wind strengthens is also
consistent with earlier studies [Thorndike and Colony, 1982]. These observed deviation angles are
comparable with their theoretical values calculated using Eq. (4). The finding that the estimated
values of the wind factor and the deviation angle are approximately within the range of typical surface
wind parameters (i.e., 2% for the wind factor and 30° for the deviation angle) in the Arctic Ocean
confirms that sea ice velocity in the ESS is controlled predominantly by wind stress drag: thus, the
influence of ocean currents is not essential.
It is interesting that the prediction skill of SIT in early summer remains at high level after the
lead time of 4 days (Fig. 6), despite the poorer prediction skill of sea ice velocity (Fig. 7a). This
suggests that the SIT prediction skill after a lead time of 4 days is not attributed to the dynamical
process but rather the thermodynamic process (i.e., the melting process of sea ice). To evaluate the
effect of sea ice melting on SIT prediction skill, we roughly estimated the thermodynamic SIT change
based on a simple sea ice melting model, as follows:
$$h^p \left( t \right) = h^a \left( t_0 \right) + \Delta t \times d\overline{h} / dt , \tag{5}$$





where $h^p$ is the predicted thermodynamic SIT change, $h_i^a$ is the initial condition, which is derived
from the analysis SIT, and $d\overline{h}/dt$ is the rate of reduction of SIT due to sea ice melting. It is known
that the summertime surface heat flux in the Pacific sector of the Arctic Ocean is dominated by the
shortwave radiation flux [Perovich et al. 2007; Steele et al. 2008]. Recently, the seasonal evolution
of sea ice retreat in early summer has been found to be explained well by a simplified ice−ocean
coupled model, in which shortwave radiation is assumed constant [Kashiwase et al. 2017]. Therefore,
as the melting rate of the SIT in each year, we used the reduction rate of SIT calculated from the
climatological analysis SIT data during 2013−2016, which is likely to reflect the typical
thermodynamic melting rate in recent years and the SIT change due to transient sea ice advection
seems to be negligible. Here, we also evaluate the prediction skill of the persistency in the initial SIT
in the ESS (first term of the RHS in Eq. (5)).

Figure 10 shows the prediction skills of early summer SIT in the simple sea ice melting and

persistency models. The prediction skill of the simple melting model, which is lower than the full
physics model, is very similar to that of the persistency model up to 3 days. However, the prediction
skill of the simple melting model is comparable with that of the full physics model after a lead time
of 4 days, which is higher than that of persistency. Figure 11 shows the temporal evolutions of SIT
difference between the forecast and analysis data in each prediction model in the period 2−9 July
2015. From the lower panel of Fig. 11, we found that the prediction skill of the full physics model is
higher than the simple melting and persistency models for lead times of 0−5 days, but comparable
with the prediction skill of the simple melting model at longer lead times (> 6 days). In the SIT
difference map of the full-physics model minus the operational analysis, a positive anomaly (i.e.,
overestimation of SIT), is evident along the sea ice edge at a lead time of 4 days, and then gradually
increases until a lead time of 8 days. For the case of the simple melting model, a similar positive
anomaly emerges at a lead time of 4 days, but the positive anomaly appears stationary along the





coastal region, compared to the full physics model. The persistency model overestimates SIT over
the entire region during the prediction. These results support the idea that the melting process is
important in the prediction of early summer SIT over longer timescales. Looking back at the seasonal
dependency of SIT prediction skill (Fig. 5), the loss of prediction skills past the 4th day in
December–February appear larger than in June–August. The difference in prediction skill between
lead times of 4 day and 9 day, averaged in January–February, is 0.05, which is somewhat larger than
in June–July (0.03). This result implies that the wintertime SIT prediction skill without any
thermodynamic melting process is largely controlled by the weak skill of atmospheric prediction, and
thus indirectly supports the assertion that the extension of the skillful prediction of early summer SIT
is attributable to the thermodynamic melting process.

**5. Case study of ice-blocked incident in the ESS in July 2014**
In the perspective of operational application of the TOPAZ4 sea ice data to the maritime
navigation of the NSR, we briefly examine the relationship between the sea ice conditions and AIS
vessel speed data for the case of an ice-blocking incident involving two vessels, based on the TOPAZ4
reanalysis data. Figure 13 shows the vessel tracks during July 4–30 2014, when the two vessels
became blocked in the ESS for about one week. During this period, SIT in excess of 100 cm is found
in the ESS with the maximum thickness of 150 cm. A joint statistical analysis of the daily mean SIT
in the TOPAZ4 reanalysis and the vessel speed along the route indicates that vessel speed is
significantly anticorrelated with SIT (−0.80) during the entire passage (Fig. 14a), significant at the
95 % confidence level based on a Monte Carlo technique [Kaplan and Glass, 1995]. The correlation
between the SIC and vessel speed is also significant (r=−0.77), although the absolute value of the
correlation coefficient is lower than for SIT. This result suggests that vessel speed was influenced by
sea ice stress due to SIT and indirectly supports the reliability of the daily mean SIT of the TOPAZ4
reanalysis data in the ESS in early summer.




## 6. Summary and discussion


In this study, the medium-range forecast skill of early summer SIT distribution in the ESS was
evaluated using the TOPAZ4 data assimilation system. Comparisons between the observed,
operational model, and TOPAZ4 reanalysis SIT data showed that the TOPAZ4 reanalysis reproduces
the observed seasonal variation (maximum in April–May and minimum in October–November)
including the rates of advance and melting of sea ice in the ESS. Earlier studies have identified that
the SIT of the TOPAZ4 reanalysis data is underestimated, even in the ESS, but the negative bias
relative to the in situ and satellite observations was about 20 cm from winter to summer, which is
smaller than another reliable hindcast model output (PIOMAS). Thus, the TOPAZ4 SIT data could
be considered reliable estimates for the ESS even in the absence of satellite observations in summer.
The prediction skill of the SIT distribution in the TOPAZ4 forecast system was examined in the
ESS using a pattern correlation analysis. Although the prediction skill was relatively lower in early
summer (June–July) with a large spread, the SIT distribution was predicted skillfully for a lead time
of up to 3 days, and the prediction skill drops abruptly after the 4th day. A similar change in prediction
skill was also found for sea ice velocity and surface wind speed over the ESS. Diagnostic analysis of
the sea ice velocity variability revealed that the early summer ice speed and direction over the EES
could be explained well by the free-drift mechanism with a wind factor of 2.2 % and a deviation angle
of 30°–50°. There results suggested that the large reduction of prediction skill could be attributed to
the process of dynamical advection of sea ice; thus, the prediction of early summer SIT distribution
will depend on precise prediction of the surface wind. Our comprehensive analysis supports an earlier
study that suggested the dynamical processes have an essential role in the prediction skill of sea ice
distribution on short timescales [Ono et al., 2016].
The time evolution of SIT and the related ice velocity relates the large difference between the
forecast and analysis data at a lead time of 4 days to the low forecast skills for an Arctic cyclone event.





Jung and Matsueda [2017] highlighted that large-scale atmospheric fluctuations in the Arctic region
in winter are predicted skillfully for lead times of up to 5 days in the operational forecast system,
which is very similar to the prediction skill in mid-latitude regions. However, Yamagami et al. [2018]
reported that the skillful prediction of Arctic cyclones generated in summer is limited to 4 days, which
is shorter than the case for the mid-latitudes [Froude, 2010]. As this area is located near the transit
zone of summertime storm tracks generated over Eurasia [Serreze and Barry, 1988], the predictability
of Arctic cyclones could be an important factor in the determination of the lead time of surface wind
speed and thus, of the SIT distribution in the ESS. The low prediction skill of the meridional wind
and ice speed suggested that the meridional component of sea ice advection contributes substantially
to the SIT distribution in the ESS. Since it was reported that additional radiosonde observations over
the Arctic Ocean have considerable impact on the prediction skill in synoptic-scale fluctuations
[Inoue et al., 2015; Yamazaki et al., 2015], additional radiosonde observations acquired over the
Arctic Ocean could lead to further extension of the lead time for medium-range forecast skill of SIT
distribution.
It is interesting that the prediction skill of early summer SIT remains at a high level after a lead
time longer than 4 days in spite of the poor prediction skill of the sea ice velocity and surface wind
fields. Based on sensitivity experiments using a simple melting and a persistency model, it was found
that the longer timescale prediction of SIT in early summer could be attributed to the thermodynamic
melting process. As the shortwave radiation flux is maximum in early summer (June–July), the
change of SIT due to the advection in relation to synoptic-scale atmospheric fluctuations is likely to
be smaller than the thermodynamic SIT reduction along the sea ice edge. Although the recognition of
the importance of the thermodynamic melting process on sea ice prediction on seasonal timescales
has been pointed out by earlier studies [Kimura et al. 2012; Bushuk et al. 2017; Kashiwase et al.
2017], our study clarified that the influence has a substantial role on the medium-range forecast of
early summer SIT distribution. Thus, the influence of sea ice advection on early summer sea ice



prediction is limited to a lead time of 4−5 days, but is dominated by the thermodynamic melting
process in later stage of the lead times. In other words, the SIT prediction skill in early summer is not
necessarily worse at the longer timescale. It is noteworthy that the dynamical process is not
unimportant for the long-term prediction in the SIT distribution in early summer, because the skillful
prediction skill at a lead time of 3 days is important as the initial conditions for the melting process
dominated for a lead time longer than 4 days. Thus, it is concluded that the atmospheric prediction
skill for a lead time of up to 3 days contributes to the short and medium-range prediction skill of the
SIT distribution in early summer.
In view of the operational application of the TOPAZ4 sea ice data to the navigation in NSR, this
study found that during an ice-blocking event that affected two tankers in the ESS in July 2014,
significant SIT (~150 cm) was simulated over the ESS by TOPAZ4. Given that the SIT is found to be
underestimated by 20 cm in TOPAZ4, the true SIT is expected to be above 150 cm. Statistical analysis
suggested that vessel speed was highly anticorrelated with the daily mean SIT variations (−0.80)
rather than the SIC (−0.77). This result demonstrated the reliability of the early summer SIT
distribution in the TOPAZ4 reanalysis data and its high potential for operational use in support of
maritime navigation of the NSR. However, this result was only based on a case study of two ships in
July 2014. To clarify the determinant factor on vessel speed, comprehensive statistical analysis will
be needed based on the speed data of different types of vessel.
Future projections for storm track activity (intensity and number) under the scenario of Arctic
climate change have been addressed by several researchers. For example, based on control
experiments using climate models, Bengtsson et al. [2006] found that summertime storm activity is
expected to increase. Orsolini and Sorteberg [2009] found that the number of storms, particularly
along the Eurasian Arctic coast, could increase in the future, because of the local enhancement of the
meridional temperature gradient between the Arctic Ocean and the warmed Eurasian continent. Nishii
et al. [2015] supported that their findings based on analyses using the Coupled Model Intercomparison



Project (CMIP) -3 and -5, although they highlighted that the CMIP projections had considerable
uncertainty. Thus, further investigations of the formation and the development mechanisms of
summertime Arctic cyclones are needed for the improvement of the prediction skill of atmospheric
wind conditions, which are responsible for the forecast skill of early summer sea ice distribution over
4 days.



**Acknowledgements**
The AMSR2 brightness temperatures and products data were provided by the Japan Aerospace
Exploration Agency (JAXA). The dataset of AMSR2 SIT and SIC was archived and provided by the
Arctic Data archive System (ADS), which was developed by the National Institute of Polar Research
(NIPR). The merging of CryoSat-2 und SMOS data was funded by the ESA project SMOS+ Sea Ice
(4000101476/10/NL/CT and 4000112022/14/I-AM) and data from 2010 to 2014 were obtained from
http://www.meereisportal.de (Grant No.: REKLIM-2013-04). The ECMWF atmospheric forecast
data were provided by the ECMWF TIGGE portal site via the TIGGE medium of the University of
Tsukuba (http://gpvjma.ccs.hpcc.jp/TIGGE/). The TOPAZ4 forecast data were analyzed using the
Pan-Okhotsk Information System of ILTS. This work was funded by the Arctic Challenge for
Sustainability (ArCS) project of the Ministry of Education, Culture, Sports, Science and Technology
in Japan. We thank James Buxton MSc from Edanz Group (www.edanzediting.com./ac) for correcting
a draft of this manuscript.



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



**Table 1.** List of observed and simulated sea ice thickness datasets

| Data sources | | Period | Spatial resolution | Time step |
|---|---|---|---|---|
| TOPAZ4 | Reanalysis | 2011–2014 | 12.5 km | Daily |
| | Forecast | 2013–2016 | 12.5 km | Daily |
| CS2SMOS | | 2011–2014 (October to April) | ~25 km | 7 days |
| IMB | | 26 March to 29 July 2014 | Point-wise | Hourly |
| PIOMAS | | 2011–2014 | ~0.8° | Daily |

**Table 2.** Pattern correlations between monthly mean climatologies of SIT in CS2SMOS, and the TOPAZ4 and PIOMAS models over the Arctic marginal seas (Laptev, East Siberian, and Chukchi Seas)

| Month | Jan. | Feb. | Mar. | Apr. | Oct. | Nov. | Dec. |
|---|---|---|---|---|---|---|---|
| TOPAZ4 | 0.97 | 0.96 | 0.93 | 0.89 | 0.97 | 0.98 | 0.97 |
| PIOMAS | 0.97 | 0.95 | 0.92 | 0.90 | 0.98 | 0.98 | 0.97 |

**Table 3.** Monthly mean SIT biases relative to observed SIT averaged over the ESS

| SIT bias (cm) | CS2SMOS (2011–2014) | | IMB (2014) | | |
|---|---|---|---|---|---|
| | Mar. | Apr. | May | Jun. | Jul. |
| TOPAZ4 | −23 | <1 | 25 | 17 | <1 |
| PIOMAS | 48 | 66 | 47 | 28 | 37 |





**Figure captions**
**Figure 1.** Spatial distribution of climatological monthly mean of SIT (cm) in April during 2011–
2014: (a) CS2SMOS, (b) TOPAZ4 reanalysis, and (c) their difference (cm). The boundaries of the
ESS and Arctic marginal seas are indicated in panel (a) by thick and thin lines, respectively.
**Figure 2.** Time series of daily mean SIT (cm) averaged over the ESS (rectangular region denoted by
black line in Fig. 1 (a)) derived from CS2SMOS (black), TOPAZ4 reanalysis (red), and PIOMAS
(blue) from January 2011 to August 2014. For CS2SMOS data, 7 day mean values are shown.
**Figure 3.** The IMB trajectory near the ESS from 26 March to 29 July 2014. (a) Spatial distribution
of daily mean SIC (%) in the AMSR2 on 29 July 2014. (b) Time series of SIT (cm) of IMB (black),
TOPAZ4 reanalysis (red), and PIOMAS (blue) along the IMB buoy trajectory (shown in panel a).
**Figure 4.** Spatial distribution of (a) monthly mean (colors) climatological SIT (m) in the TOPAZ4
reanalysis and (b) the RMS variability of daily mean SIT (colors) in July during 2011–2014. The
monthly mean of climatological SIC (white contours) in July is indicated in panel (a). The rectangular
region enclosing the ESS (70°–80°N, 150°–180°E) is shown in panel (b). (c) Time series of monthly
mean SIT (grey shade) and RMS of TOPAZ4 reanalysis (black line) averaged over the ESS. The
scale of the RMS is indicated on the right axis.
**Figure 5.** The PCCs (colors) between operational forecast and analysis SIT in the ESS (70°–80°N,
150°–180°E) in each month, averaged from 2013–2016. The isoline of standard deviation of the
PCCs at 0.05 is shown with white contours.
**Figure 6.** PCCs between forecast and analysis SIT from operational TOPAZ4 data in early summer
(June–July) averaged on 2013–2016. Error bar indicates the standard deviation of the PCCs.
**Figure 7.** The PCCs between forecast and analysis (a) zonal (black) and meridional ice speed (red)
and (b) zonal (black) and meridional (red) surface wind speed in June–July averaged from 2013–2016.
Error bar indicates the standard deviation of the PCCs.



**Figure 8.** Temporal evolution of SIT (cm; colors) and ice velocity (m s$^{-1}$; vectors) distribution for
(left) analysis, (center) forecast, and (right) the difference between forecast and analysis at increasing
lead times from +0 day to +6 days initialized on 2nd July 2015. The corresponding PCCs for the SIT
(black), zonal (red) and meridional ice speeds (blue) in the ESS (right-lower panel of the time
evolution) are shown in the lower panel. The scale for the PCCs of the zonal and meridional ice
speeds is indicated on the right axis.
**Figure 9.** (a) Relationship between 10m wind speed (m s$^{-1}$) in the ERA Interim reanalysis data and
sea ice speed (m s$^{-1}$) in the TOPAZ4 reanalysis averaged over a part of the ESS (72°–76° N,
150°–170° E) during 1–31 July 2011–2014. Broken and solid lines indicate the regression line of ice
speed on 10m wind speed ( $y = 0.0224x - 0.0112$ ) and the theoretical ice speed estimated based on
classical free-drift theory, respectively. (b) Angle (degrees) of sea ice velocity relative to surface wind
vectors averaged over the ESS. Positive values indicate sea ice drift is to the right of the wind direction.
Solid curve indicates the wind–ice velocity angle estimated based on classical free-drift theory.
**Figure 10.** The PCCs between forecast and analysis SIT from the full physics model (black),
persistency (red), and a simple melting model (blue) in July averaged from 2013–2016. Error bar
indicates the standard deviation of the PCCs.
**Figure 11.** Temporal evolution of SIT differences (cm; colors) between the forecast and analysis data
at lead times increasing from +2 to +8 days, initialized on 2nd July 2015. In each panel, the sea ice
edge of the analysis, defined by 30% SIC, is shown. Corresponding PCCs for the full physics model
(black), a simple melting model (red) and persistency (blue) in the ESS (right-lower panel of the time
evolution) are shown in the lower panel.
**Figure 12.** Trajectory of the two tankers over the ESS based on AIS data. The routes cross the ESS
from the Laptev Sea on 4 July 2014 to the port of Yamal on 31 July 2014, via the port of Pevek on
20 July 2014. The forward route is highlighted by green circles. The SIT (cm; colors) and SIC (%;
contours) averaged over the period of the forward route are shown**.**





**Figure 13.** Scatter plots of daily mean vessel speeds (knots) and sea ice thickness (cm) from 4–30
July 2014.






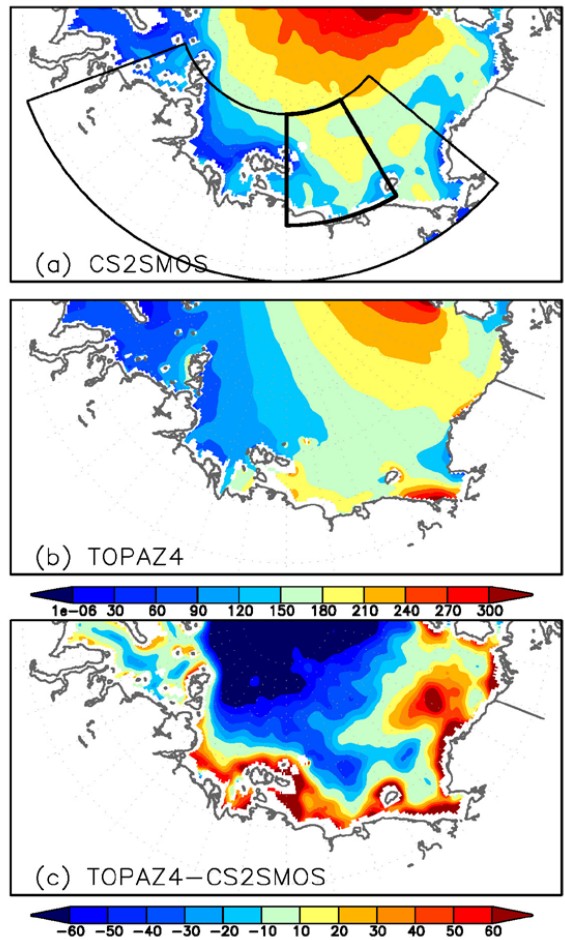


**Figure 1.** Spatial distribution of climatological monthly mean of SIT (cm) in April during 2011–

2014: (a) CS2SMOS, (b) TOPAZ4 reanalysis, and (c) their difference (cm). The boundaries of the

ESS and Arctic marginal seas are indicated in panel (a) by thick and thin lines, respectively.





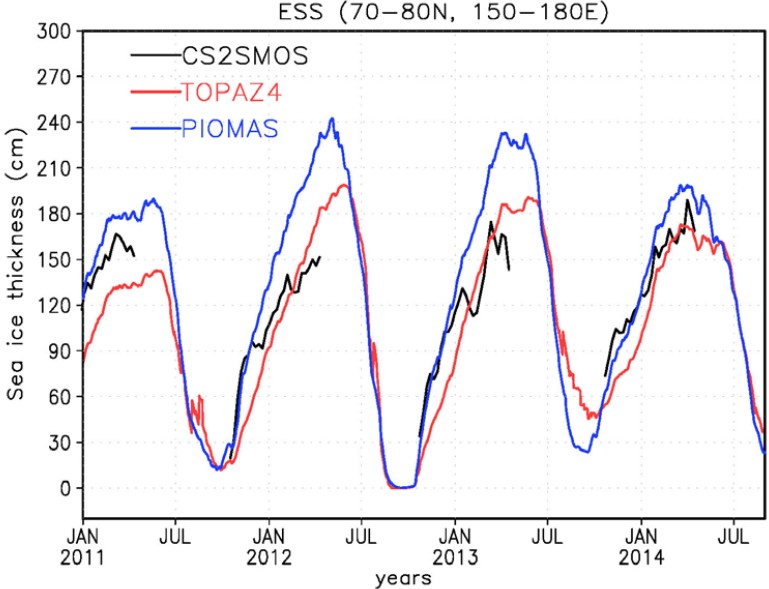


**Figure 2.** Time series of daily mean SIT (cm) averaged over the ESS (rectangular region denoted by

black line in Fig. 1 (a)) derived from CS2SMOS (black), TOPAZ4 reanalysis (red), and PIOMAS

(blue) from January 2011 to August 2014. For CS2SMOS data, 7 day mean values are shown.



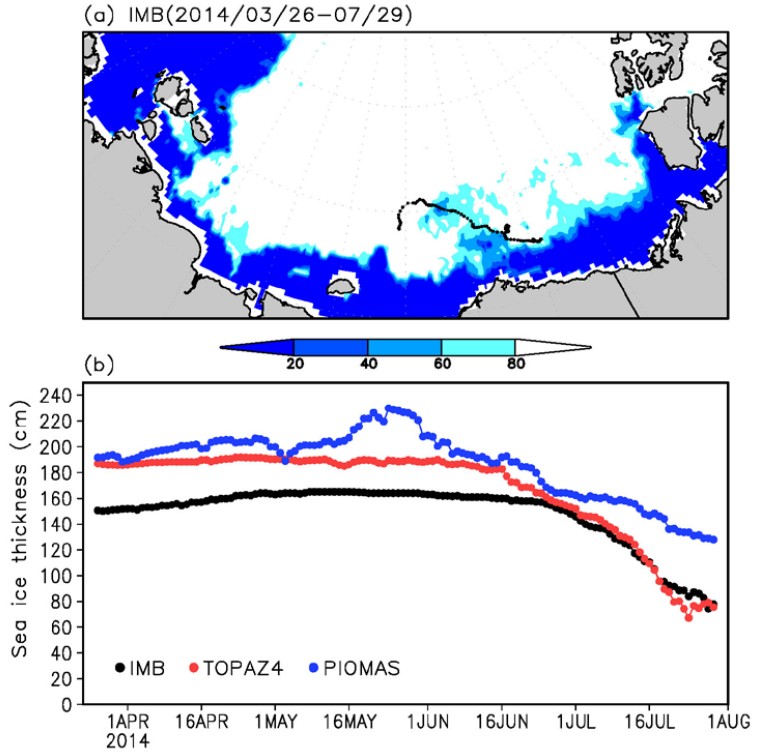


**Figure 3.** The IMB trajectory near the ESS from 26 March to 29 July 2014. (a) Spatial distribution

of daily mean SIC (%) in the AMSR2 on 29 July 2014. (b) Time series of SIT (cm) of IMB (black),

TOPAZ4 reanalysis (red), and PIOMAS (blue) along the IMB buoy trajectory (shown in panel a).





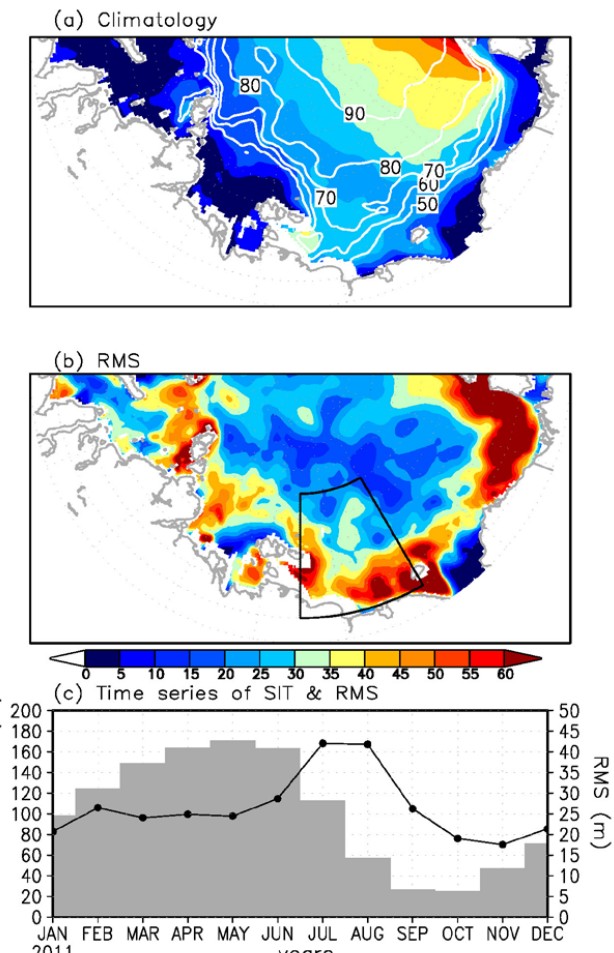


**Figure 4.** Spatial distribution of (a) monthly mean (colors) climatological SIT (m) in the TOPAZ4

reanalysis and (b) the RMS variability of daily mean SIT (colors) in July during 2011–2014. The

monthly mean of climatological SIC (white contours) in July is indicated in panel (a). The rectangular

region enclosing the ESS (70°−80°N, 150°−180°E) is shown in panel (b). (c) Time series of monthly

mean SIT (grey shade) and RMS of TOPAZ4 reanalysis (black line) averaged over the ESS. The

scale of the RMS is indicated on the right axis.





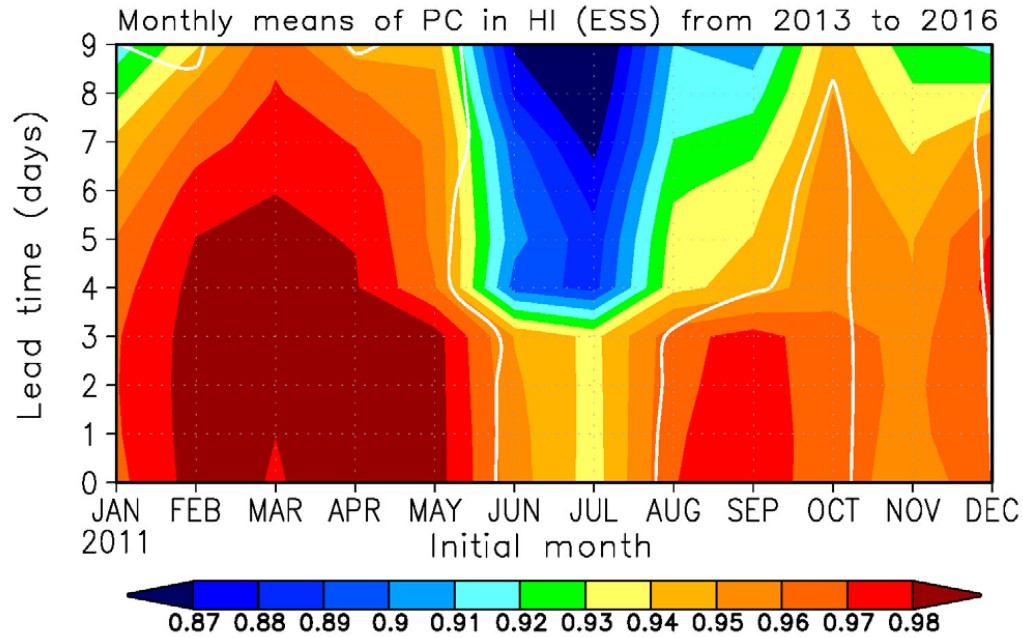

**Figure 5.** The PCCs (colors) between operational forecast and analysis SIT in the ESS (70°−80°N,
150°−180°E) in each month, averaged from 2013−2016. The isoline of standard deviation of the
PCCs at 0.05 is shown with white contours.




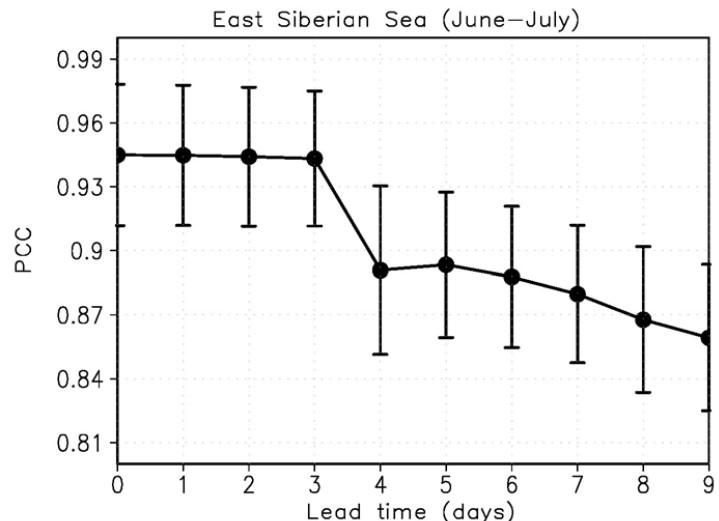

729

**Figure 6.** PCCs between forecast and analysis SIT from operational TOPAZ4 data in early summer

(June–July) averaged on 2013–2016. Error bar indicates the standard deviation of the PCCs.

732



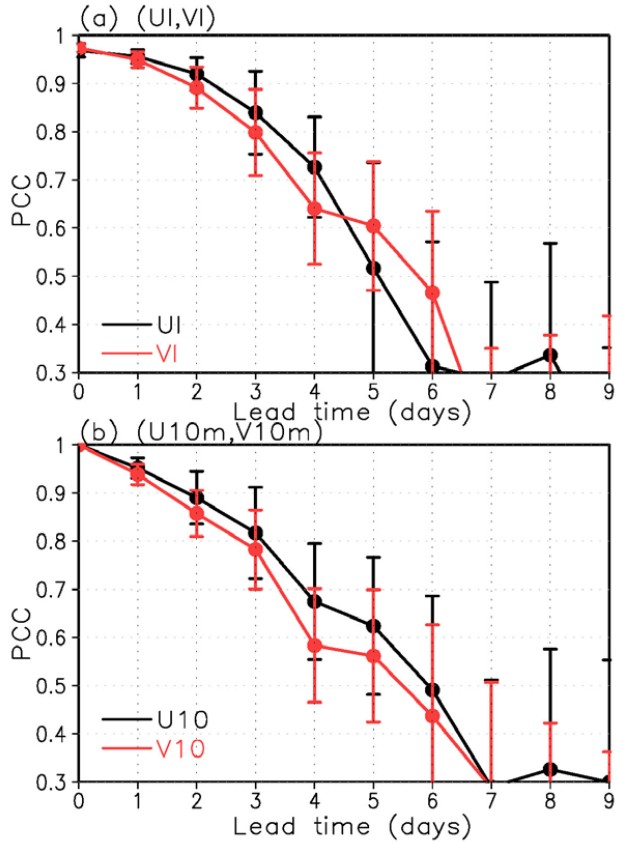

733

**Figure 7.** The PCCs between forecast and analysis (a) zonal (black) and meridional ice speed (red)

and (b) zonal (black) and meridional (red) surface wind speed in June–July averaged from 2013–2016.

Error bar indicates the standard deviation of the PCCs.





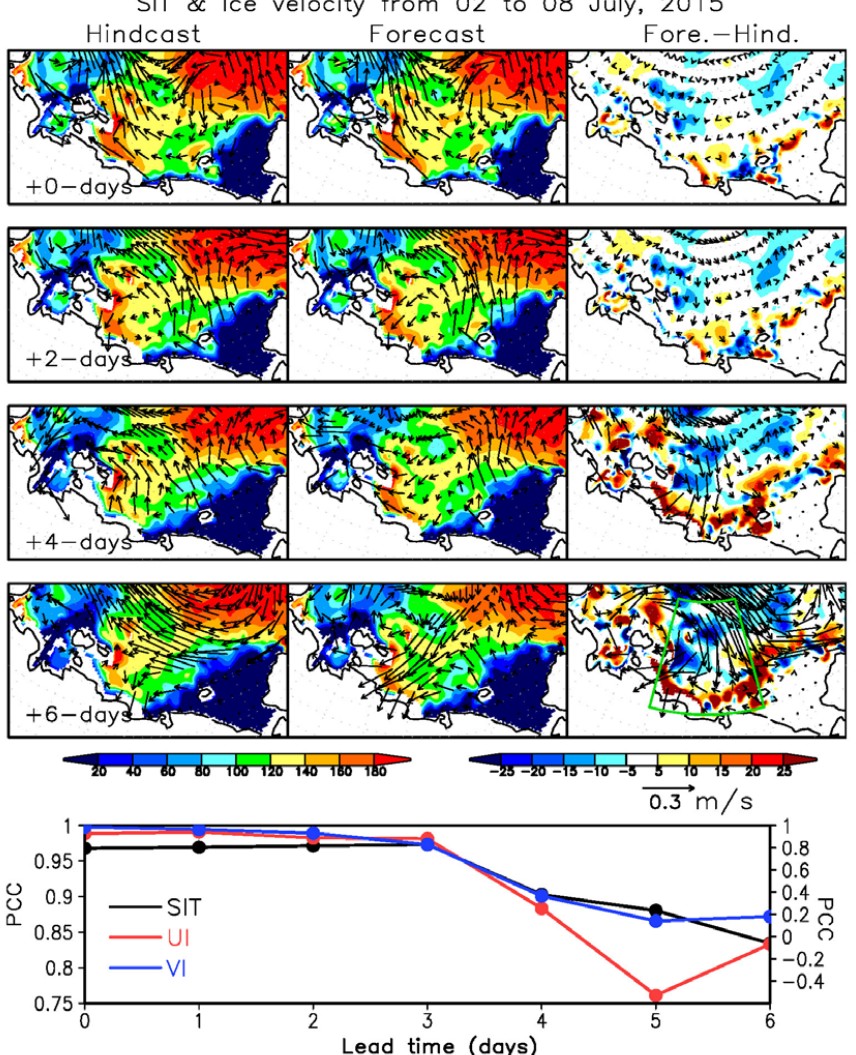

**Figure 8.** Temporal evolution of SIT (cm; colors) and ice velocity (m s$^{-1}$; vectors) distribution for
(left) analysis, (center) forecast, and (right) the difference between forecast and analysis at increasing
lead times from +0 day to +6 days initialized on 2nd July 2015. The corresponding PCCs for the SIT
(black), zonal (red) and meridional ice speeds (blue) in the ESS (right-lower panel of the time
evolution) are shown in the lower panel. The scale for the PCCs of the zonal and meridional ice
speeds is indicated on the right axis.





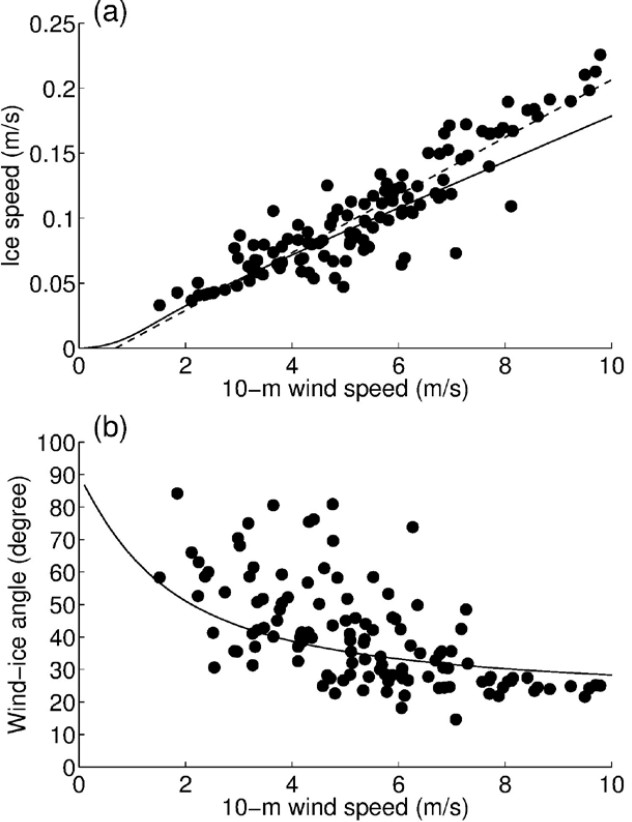

744

**Figure 9.** (a) Relationship between 10m wind speed (m s$^{-1}$) in the ERA Interim reanalysis data and

sea ice speed (m s$^{-1}$) in the TOPAZ4 reanalysis averaged over a part of the ESS (72°–76° N,

150°–170° E) during 1–31 July 2011–2014. Broken and solid lines indicate the regression line of ice

speed on 10m wind speed ( $y = 0.0224x - 0.0112$ ) and the theoretical ice speed estimated based on

classical free-drift theory, respectively. (b) Angle (degrees) of sea ice velocity relative to surface wind

vectors averaged over the ESS. Positive values indicate sea ice drift is to the right of the wind direction.

Solid curve indicates the wind–ice velocity angle estimated based on classical free-drift theory.

752



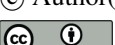

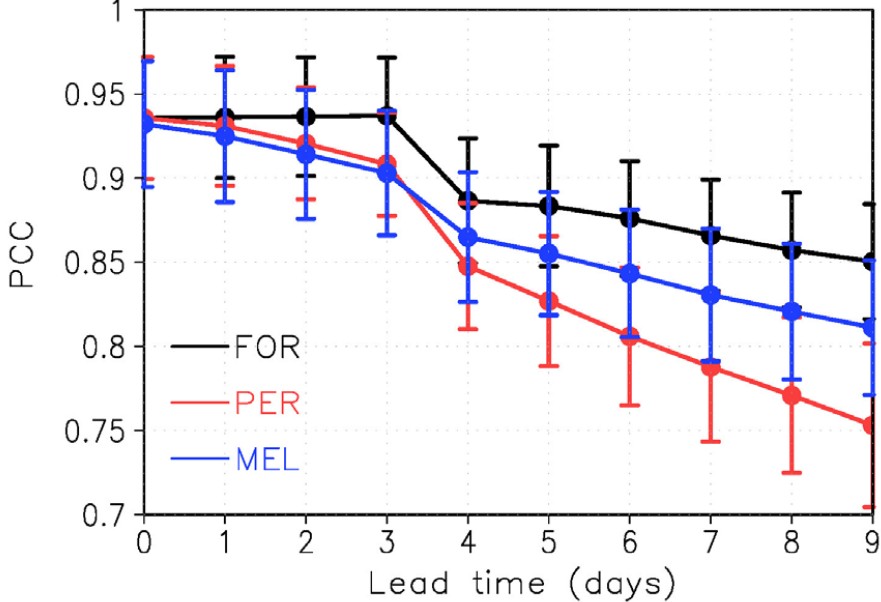

753

**Figure 10.** The PCCs between forecast and analysis SIT from the full physics model (black),

persistency (red), and a simple melting model (blue) in July averaged from 2013–2016. Error bar

indicates the standard deviation of the PCCs.

757





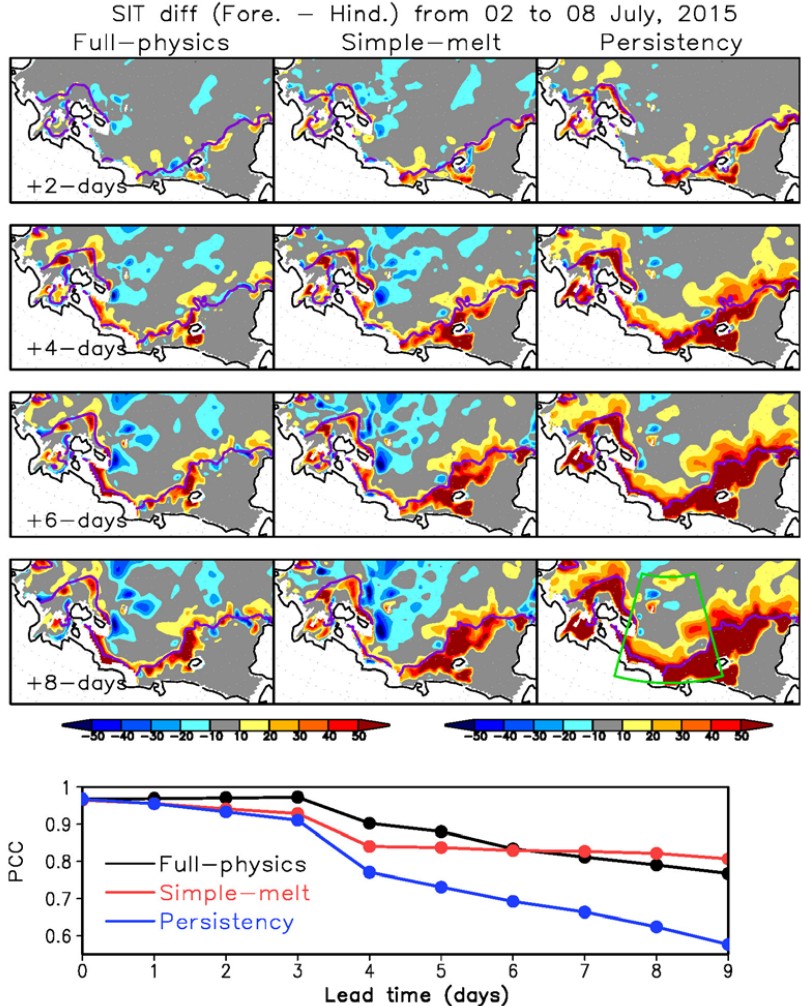

**Figure 11.** Temporal evolution of SIT differences (cm; colors) between the forecast and analysis data at lead times increasing from +2 to +8 days, initialized on 2nd July 2015. In each panel, the sea ice edge of the analysis, defined by 30% SIC, is shown. Corresponding PCCs for the full physics model (black), a simple melting model (red) and persistency (blue) in the ESS (right-lower panel of the time evolution) are shown in the lower panel.



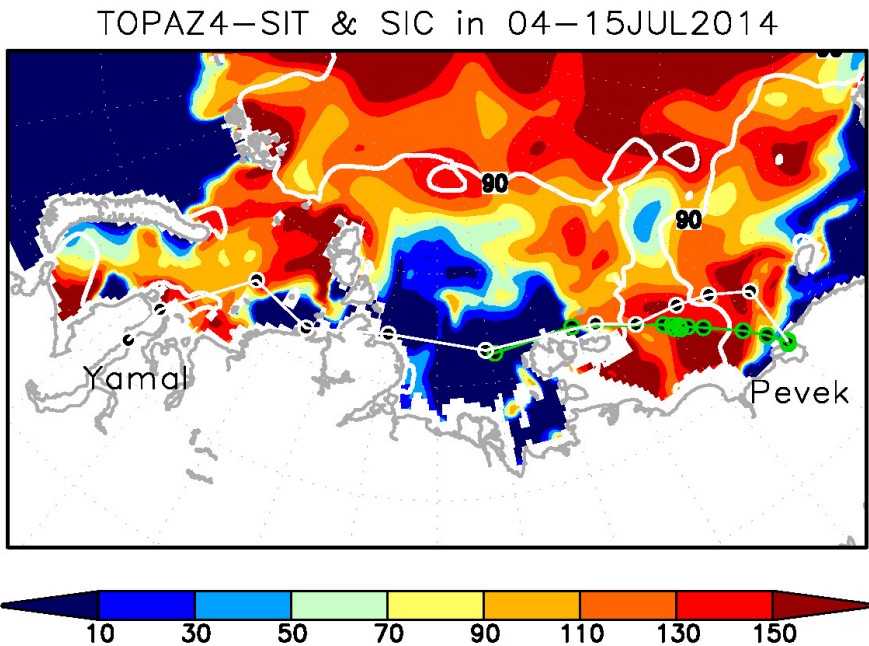

**Figure 12.** Trajectory of the two tankers over the ESS based on AIS data. The routes cross the ESS

from the Laptev Sea on 4 July 2014 to the port of Yamal on 31 July 2014, via the port of Pevek on

20 July 2014. The forward route is highlighted by green circles. The SIT (cm; colors) and SIC (%;

contours) averaged over the period of the forward route are shown**.**





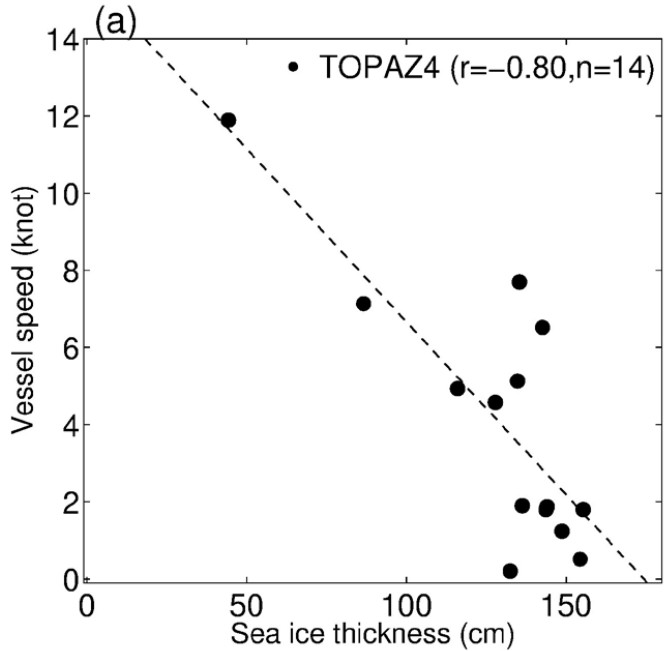

770

**Figure 13.** Scatter plots of daily mean vessel speeds (knots) and sea ice thickness (cm) from 4–30

July 2014.