# Peer review of "Medium-range predictability of early summer sea ice thickness distribution in the East Siberian Sea: Importance of dynamical and thermodynamic melting processes"

_The Cryosphere, 2018_

## Referee Comment (RC1) · Anonymous Referee #1 · 11 Mar 2018

General Comments

The paper addresses a relevant and current topic, seasonal sea ice prediction as it pertains to increased maritime operations in northern waters in the summer season. The authors highlight the utility of the TOPAZ4 forecast system for estimating sea ice thickness distributions in the East Siberian Sea, an area that has seen increased vessel activity during summer in recent years. Sea ice thickness outputs are compared to satellite (Cryosat-2 and SMOS) and in situ (ice mass balance buoy) observations, with a negative bias of 20cm from winter to summer shown to be smaller than other model

outputs. Skillfull predictions of sea ice thickness are limited to lead times of up to 3 days due to the influence of dynamical processes, which is somewhat expected based on similar studies and here attributed to the influence of Arctic cyclones on sea ice drift. Interestingly, the authors study the effect of thermodynamic melting processes on sea ice thickness prediction skill at longer time scales, demonstrating dependency of prediction skill on those processes. A case study of two ships is used to show how vessel speeds were related to TOPAZ4 sea ice thickness estimates in July, when ice thickness up to 150cm caused vessel blocking.

The paper is well written, the data and methods generally well described, and the results presented and discussed in a logical manner with clear figures and tables. Descriptions of data and methods are clear enough to allow repeatability.

Some further editing is needed (e.g. reference to Fig. 14 on Line 370; "There" instead of "Their" on Line 394) but otherwise there isn't any need to make any major adjustments to the text like removing or combing sections.

Specific Comments

The title of the paper is perhaps too broad given that the focus is on the performance of the TOPAZ4 system on predictions in the East Siberian Sea, rather than an overall assessment of dynamic and thermodynamic processes on medium-range predictions.

The authors use the merged Cryosat-2/SMOS satellite-based sea ice thickness product to evaluate TOPAZ4 sea ice thickness estimates. Some qualitative statements about the uncertainty of this product are made, but more information on potential bias is needed since these data are used to assess TOPAZ4 (and PIOMAS) outputs (see Figure 2).

The authors need to be cautious about attributing model skill from a comparison between simulated sea ice thickness and limited measurements from ice mass balance buoys in a single melting season (2014). The agreement is certainly good, but the

statement made on Lines 226-230 is not well supported given the lack of supporting data. If more comparisons are possible, they would certainly add value to the paper.

---

## Referee Comment (RC2) · Anonymous Referee #2 · 15 Mar 2018

General comments:

This paper evaluates (1) sea ice thickness (SIT) from the 4th version of the Towards an Operational Prediction system for the North Atlantic European coastal Zones (TOPAZ4) ocean data assimilation system and (2) medium range forecast of SIT distribution in the Eastern Siberian Sea (ESS) from the TOPAZ ocean data assimilation system forced by the ECMWF atmospheric medium-range forecast data. The evaluation of TOPAZ4 SIT uses observational data from satellite retrievals, in situ observations and model generated output from the Pan-Arctic Ice Ocean Modeling and Assimilation System

(PIOMAS). The forecast evaluation analyzes impacts of dynamic and thermodynamic processes. Descriptions of the methods and analysis are clear. The results are interesting. I recommend the paper be accepted for publication after a minor revision.

Specific comments

1. The SIT from the TOPAZ4 assimilation contains large errors which are comparable to that in PIOMAS. I would suggest that the both TOPAZ4 SIT and PIOMAS SIT be used for the evaluation of the forecast to reduce the observational uncertainties. I also suggest PIOMAS be included in Figure 1.

2. A large portion of the PCC skill in Figure 5 is from the persistence. A comparison with persistence skill is needed to see to what extent the sill in Figure 5 has benefited from the persistence of the initial anomalies.

3. Lines 134-138. Move the portion "In this . . .process [Startk et al. 2008]" into the first paragraph of section 2.

4. Line 145. How is the 10-member ensemble produced?

5. Line 146. Spell out ECMWF.

6. Line 149. Please make clear how the 259 cases come out.

7. Line 172. Spell out PIOMAS.

8. Line 272. Change "completely" to "largely". The correlation shows that they are still related to some extent.

9. Line 371. Fig.14a does not exist.

10. "Figure" and "Fig." are used interchangeably.

---

## Referee Comment (RC3) · Anonymous Referee #3 · 19 Mar 2018

This manuscript investigates the forecast skill of the sea ice thickness distribution in the East Siberian Sea in early summer for a lead time from a few days to 10 days. The description and validation of the TOPAZ4 reanalysis utilized for this analysis are clear. They demonstrate the characteristic time evolution of the prediction skill and suggest the reasons for such as the abrupt reduction of the skill after 4 days. Their explanations by using simple models are reasonable and useful for the community of the Arctic sea ice monitoring and prediction. Therefore, I recommend this manuscript to be accepted for publication in the Cryosphere.

[Figure]

Please check the minor comments as described below.

L53: "CMIP" firstly appears here.

L212: The bias quantities described in this paragraph seem to depend on the definition of the ESS with negative biases in the north and positive biases in the south (Fig. 1c). This should be mentioned here and the conclusion section. I believe it does not degrade the analysis of this study and help to avoid too much damaging the pedigree PIOMAS data.

L230: I consider that the 2nd "errors" can be eliminated.

L252: The abrupt reduction in October in Fig. 5 is not clear to me. Please chick this.

L261: Please weaken the statement "the SIT distribution has a zonally homogeneous pattern".

L272: "directed southward" should also be weakened if it is not exactly southward.

L277: "a deficiency at predicting Arctic cyclone". Please check this "at". (I am not a native English speaker and sorry if this is correct.)

L312 and L313: I think the units "cm s-1" should be "m s-1". Please check them.

L320: Since the authors describe on the reduction of the prediction skill in the 4th day, some words should be added to "remains at high level after the lead time of 4 days" on how high it is in order to avoid confusing.

L358: "controlled by the weak skill of atmospheric prediction" is not clear to me.

L366: "Figure 13" -> "Figure 12"

L370: "Fig. 14a" -> "Fig. 13"

L371: Please provide the significance of the difference between these two correlation coefficients if possible. Even if it is not significant, please do not consider to delete this interesting section. The sample number will be increased in the future to determine its

significance as described in the final section. However, more careful discussion seems to be required for the conclusion in this section, since 1) the SIT and SIC time series can be resemble and 2) reproduction of SIC in the TOPAZ4 reanalysis is not validated in this study.

---

## Author Comment (AC1) · 24 Apr 2018

Dr. John Yackel Editor, The Cryosphere

Dear Dr. John Yackel

Thank you very much for your treatment of the article entitled "Medium-range predictability of early summer sea ice thickness distribution in the East Siberian Sea: Importance of dynamical and thermodynamic melting processes" by Takuya Nakanowatari and co-authors (tc-2018-25). We have read the comments given by

the reviewers with great interest and want to express my thanks to them through you. Their comments are very helpful for revising the manuscript. Since we have revised our manuscript according to the reviewer's comments thoroughly, we resubmit our manuscript.

I sent the following files in Editorial Manager, in addition to this cover letter file. 1. A response to Reviewer #1 (res1-tc2018-25.pdf) :(PDF file) 2. A response to Reviewer #2 (res2-tc2018-25.pdf) :(PDF file) 3. A response to Reviewer #3 (res3-tc2018-25.pdf) :(PDF file) 4. A copy of the manuscript with the changes noted (tc-2018-25R_noted.pdf) :(PDF file)

Note that in "Response to Reviewer Files" the pages and lines are those of "A copy of the manuscript with the changes noted". Please refer to "A copy of the manuscript with the changes noted".

We hope the manuscript has been improved satisfactorily for publication in The Cryosphere.

Sincerely yours,

Takuya Nakanowatari (nakanowatari.takuya@nipr.ac.jp) Arctic Environment Research Center National Institute of Polar Research 10-3, Midori-cho, Tachikawa-shi, Tokyo, 190-8518, Japan Tel (81)-42-512-0763

Please also note the supplement to this comment:
https://www.the-cryosphere-discuss.net/tc-2018-25/tc-2018-25-AC1-supplement.pdf

**Supplement:**

[revised manuscript text omitted]

---

## Author Comment (AC2) · 24 Apr 2018

Thank you very much for your careful reviewing of our manuscript. We found reviewer's comments most helpful and have revised the manuscript accordingly. Note that the line numbers are those of "the manuscript with changes noted (tc-2018-25R_noted.pdf)", not of "the original revised manuscript (tc-2018-25R.pdf)". Please refer to "the manuscript with changes noted".

Please also note the supplement to this comment:

[Figure]

https://www.the-cryosphere-discuss.net/tc-2018-25/tc-2018-25-AC2-supplement.pdf

[Figure]

**Supplement:**

*General Comments*
*The paper addresses a relevant and current topic, seasonal sea ice prediction as it pertains to increased maritime operations in northern waters in the summer season. The authors highlight the utility of the TOPAZ4 forecast system for estimating sea ice thickness distributions in the East Siberian Sea, an area that has seen increased vessel activity during summer in recent years. Sea ice thickness outputs are compared to satellite (Cryosat-2 and SMOS) and in situ (ice mass balance buoy) observations, with a negative bias of 20cm from winter to summer shown to be smaller than other model outputs. Skillfull predictions of sea ice thickness are limited to lead times of up to 3 days due to the influence of dynamical processes, which is somewhat expected based on similar studies and here attributed to the influence of Arctic cyclones on sea ice drift. Interestingly, the authors study the effect of thermodynamic melting processes on sea ice thickness prediction skill at longer time scales, demonstrating dependency of prediction skill on those processes. A case study of two ships is used to show how vessel speeds were related to TOPAZ4 sea ice thickness estimates in July, when ice thickness up to 150cm caused vessel blocking. The paper is well written, the data and methods generally well described, and the results presented and discussed in a logical manner with clear figures and tables. Descriptions of data and methods are clear enough to allow repeatability. Some further editing is needed (e.g. reference to Fig. 14 on Line 370; "There" instead of "Their" on Line 394) but otherwise there isn't any need to*

*make any major adjustments to the text like removing or combing sections.*
**\* We have revised the above editing errors (Line 455, 458, and 504).**

*Specific Comments*
*The title of the paper is perhaps too broad given that the focus is on the performance of the TOPAZ4 system on predictions in the East Siberian Sea, rather than an overall assessment of dynamic and thermodynamic processes on medium-range predictions.*
**\* As the reviewer pointed out, the original title may give a general aspect of the physical mechanism of SIT prediction in ESS. Since our study highly depends on the TOPAZ4 system, we have modified the title as follows;**
**"Medium-range predictability of early summer sea ice thickness distribution in the East Siberian Sea based on the TOPAZ4 ice-ocean data assimilation system"** **(Lines 5-6)**

*The authors use the merged Cryosat-2/SMOS satellite-based sea ice thickness product to evaluate TOPAZ4 sea ice thickness estimates. Some qualitative statements about the uncertainty of this product are made, but more information on potential bias is needed since these data are used to assess TOPAZ4 (and PIOMAS) outputs (see Figure 2).*
**\* Thank you very much for your notification that we have overlooked the reliability of CS2SMOS data. Indeed, it was reported that this dataset has non-negligible negative bias and errors by comparing it with the independent sea ice thickness data derived from Airborne EM sensor in the corresponding paper (Ricker et al. 2017). Therefore, we had to evaluate and discuss the reliability of the CS2SMOS in the ESS.**

**Since the CS2SMOS highly depends to the reliability of the merging SIT data, which are CryoSat-2 and SMOS SIT products [Ricker et al. 2017], there is possibility that the CS2SMOS SIT is underestimated in the ESS. To check this possibility, we briefly examined the ice type data which were used for the determination of merged SIT products. In the first ice periods from 2011 to 2013, the uncertainty of CS2SMOS SIT is out of range for that of PIOMAS, but the CS2SMOS SIT is comparable to that for PIOMAS in 2014 when the sea ice is classified as multi-year ice (Fig. 3). This result implies that the CS2SMOS SIT is underestimated in the ESS. These descriptions have been added in the revised version as follows;**
**"In the freezing season, the TOPAZ4 SIT in the ESS tends to be thinner than the PIOMAS SIT, and seems comparable to the CS2SMOS SIT. The monthly mean**

bias of TOPAZ4 SIT relative to CS2SMOS SIT is -23 cm and 1 cm in March and April, respectively (Table 3). On the other hand, we should pay attention to the possibility that the CS2SMOS SIT may be underestimated in this region, because the CS2SMOS highly depends on the reliability of merging two SIT data, which are CryoSat-2 and SMOS SIT products [Ricker et al. 2017]. To check the possibility that the CS2SMOS SIT has a negative bias in this area, we briefly examined the ice type data which were used for the determination of merged SIT products. In the period from 2011 to 2013, the uncertainty of CS2SMOS SIT is out of range for that of PIOMAS, but the CS2SMOS SIT is comparable to that for PIOMAS in 2014 when the sea ice is classified as multi-year ice (Fig. 3). This result implies that the CS2SMOS SIT is underestimated in the ESS due to the large fraction of SMOS SIT products even in the sea ice thicker than 1 m." (Lines 269-280)

In addition to this revision, the bias and uncertainty of SIT in TOPAZ4 highly depends on the data source of SIT to be used for the comparison as well as the region within the ESS. Thus, we realize that the specific value of the bias of TOPAZ4 should not be included in the abstract. Thus, we also have modified the corresponding sentence in the abstract and summary as follows;
"Comparison of the operational model SIT data to reliable SIT estimates (hindcast, satellite, and in situ data) showed that the TOPAZ4 reanalysis reproduces qualitatively the tongue-like distribution of SIT in ESS in early summer and the seasonal variations." (Lines 24-27)

"Comparisons between the operational model, observed, and TOPAZ4 reanalysis SIT data showed that the TOPAZ4 reanalysis qualitatively reproduces the tongue-like distribution of SIT in the ESS in early summer, and its seasonal variation (maximum in April–May and minimum in October–November) including the rates of advance and melting of sea ice in the ESS). Although in this region, the inherent negative bias of SIT in TOPAZ4 is relatively large in March to May, the bias is reduced in early summer (June-July) within ~±20 cm due to the excess of SIT along the coastal region in the ESS. The TOPAZ4 SIT data also shows a good correspondence with IMB buoy data in and around the ESS with the mean bias of ~9 cm and the root mean square error of ~30 cm. Thus, the TOPAZ4 SIT data could be considered reliable estimates for the ESS even in the absence of satellite observations in summer." (Lines 470-484)

*The authors need to be cautious about attributing model skill from a comparison between simulated sea ice thickness and limited measurements from ice mass balance buoys in a single melting season (2014). The agreement is certainly good, but the statement made on Lines 226-230 is not well supported given the lack of supporting data. If more comparisons are possible, they would certainly add value to the paper.*

**\* As the reviewer pointed out, only one buoy data is not enough to support the reliability of SIT in TOPAZ4. We re-checked all of the IMB data in and around the ESS and found that additional 3 buoy data are available near the ESS (Please refer Fig. 1a and Table 1 for the location and periods). Although the location of these buoy data does not necessarily cover the ESS on which we focused in this study, these data seem to be appropriate for our purpose, because the range of the climatological SIT in these region is similar to that in the ESS (Fig. 1a). The direct comparison between the TOPAZ4 and IMB shows that the mean bias and root mean square error of TOPAZ4 is 8.3 cm and 30 cm, respectively. In particular, the TOPAZ4 SIT data shows a good correspondence with IMB buoy data in 2014, which is near the ESS in July (Fig. 1a and Table 1). These results support that the reliability of TOPAZ4 SIT data in the ESS in early summer. There results and Fig. A1 have been added in the revised version (Lines 282-293 and Fig. 4).**

[Figure]

**Figure A1.** The comparisons of the daily mean SITs derived from IMB buoy data with the corresponding SIT in TOPAZ4 reanalysis data from 2011 to 2014 in and around the ESS. The SIT data are re-sampled per 7 days. The reference unit line and the regression lines onto IMB buoy data are shown by solid and dashed lines, respectively.

  According to this revision, we have removed Fig. 3 in the original version and the related sentence.

In addition to the revision based on the reviewer's comments, we also have revised the following items listed below;

1)  We have refined several sentences for clarification (e.g., Lines 146, 149, 186, 213).

2)  We removed the citation [Nakanowatari et al. 2017], which is it is a proceeding of Monbetu-2017 Symposium (Line 100) and the reference which is not cited in this paper [Nakanowatari et al. 2014] (Lines 667-669).

3)  We have updated the following reference information.

Yamagami A., Matsueda M., & Tanaka H. L. 2018. Predictability of the 2012 great Arctic cyclone on medium-range timescales, 15, 13-23, doi: 10.1016/j.polar.2018.01.002. (Lines 747-748)

---

## Author Comment (AC3) · 24 Apr 2018

Thank you very much for your careful reviewing of our manuscript. We found reviewer's comments most helpful and have revised the manuscript accordingly. Note that the line numbers are those of "the manuscript with changes noted (tc-2018-25R_noted.pdf)".

Please also note the supplement to this comment:
https://www.the-cryosphere-discuss.net/tc-2018-25/tc-2018-25-AC3-supplement.pdf

[Figure]

[Figure]

**Supplement:**

*General comments:*
*This paper evaluates (1) sea ice thickness (SIT) from the 4th version of the Towards an Operational Prediction system for the North Atlantic European coastal Zones (TOPAZ4) ocean data assimilation system and (2) medium range forecast of SIT distribution in the Eastern Siberian Sea (ESS) from the TOPAZ ocean data assimilation system forced by the ECMWF atmospheric medium-range forecast data. The evaluation of TOPAZ4 SIT uses observational data from satellite retrievals, in situ observations and model generated output from the Pan-Arctic Ice Ocean Modeling and Assimilation System (PIOMAS). The forecast evaluation analyzes impacts of dynamic and thermodynamic processes. Descriptions of the methods and analysis are clear. The results are interesting. I recommend the paper be accepted for publication after a minor revision.*

*Specific comments*
*1. The SIT from the TOPAZ4 assimilation contains large errors which are comparable to that in PIOMAS. I would suggest that the both TOPAZ4 SIT and PIOMAS SIT be used for the evaluation of the forecast to reduce the observational uncertainties. I also suggest PIOMAS be included in Figure 1.*
**\*According to the reviewer's comment, we examined the forecast skill of TOPAZ4 assuming that the PIOMAS SIT is the true value. However, the prediction skill is**

quite low in a whole season (Fig. A1). This is probably due to the spatial distribution of SIT in TOPAZ4 analysis is different from that in PIOMAS on daily mean field. Although the overall pattern of the SIT distribution in TOPAZ4 is similar with that in PIOMAS in and around the ESS in the climatological field (Fig. 1), the location of ice edge and small-scale undulation near the shelf region of ESS highly depends to the original model resolution. Since our study focus on the small-scale disturbance of SIT in the ESS, we believe that the evaluation of the prediction skill based on TOPAZ4 analysis and its forecast is appropriate for our purpose.

[Figure]

Figure A1. The prediction skill (PCC) of SIT forecast in the ESS (70°–80°N, 150°–180°E) in each month obtained from TOPAZ4 operational forecast model with PIOMAS hindcast SIT data, averaged from 2013–2016. The standard deviations of the PCCs are shown with white contours.

On the other hand, we have added the climatological SIT of PIOMAS in July in the revised version (Fig. 1) to evaluate the overall distribution of SIT in TOPAZ4 analysis. The PIOMAS show relatively thick ice (>1.0 m) extends from the North Pole to the ESS (Fig. 1a). These features are qualitatively simulated in the TOPAZ4 reanalysis data (Fig. 1b). The PCC of the climatological SIT between TOPAZ4 and PIOMAS in the Arctic marginal seas is larger than 0.9 from March

to July (Table 2). Notes that the region for the Arctic marginal seas is partly shrunk in the revised version, because we don't focus on the Kara Sea (Fig. 1a). The PCCs of the climatological SIT between TOPAZ4 and CS2SMOS from March to April are comparable to those for PIOMAS (Table 2), and thus these results support the reliability of the spatial distribution of SIT in and around the ESS. These results have been added in the revised version (Lines 225-241).

In addition, the monthly mean biases of TOPAZ4 SIT data relative to PIOMAS in Jun to July are smaller than those in March to May (Table 3), although the TOPAZ4 SIT in the ESS tends to be thinner than the PIOMAS SIT in freezing season. Also, the TOPAZ4 SIT is within the standard deviation of PIOMAS SIT anomaly in each grid relative to the area-averaged value in early summer (June-July) (Fig. 3). Thus, at least the overall spatial distribution of SIT in the ESS is qualitatively simulated in the TOPAZ4 and the inherent negative bias is suppressed in early summer, which is partly related to the compensation by the positive bias near the shelf region of the ESS. These results and discussions have been added in the revised version (Lines 242-268).

Along with this revision, Figure for the comparison of climatological SIT distribution between CS2SMOS and TOPAZ4 in April has been moved to Fig. 2.

*2. A large portion of the PCC skill in Figure 5 is from the persistence. A comparison with persistence skill is needed to see to what extent the sill in Figure 5 has benefited from the persistence of the initial anomalies.*

\* According to the reviewer's comment, we have added the prediction skill obtained from the persistency in Fig. 6b and the difference in the prediction skill between the operational forecast model and persistency (Fig. 6c). As expected, a large portion of the prediction skill originates from the persistency at the lead times of 0-3 days (the explained variance is about 95%). On the other hand, the fraction of the prediction skill related to the operational model increases at longer lead times in a whole season except for May and October. In July, the contribution of the operational model on the prediction skill reaches ~15% at 7 day lead time. These results and implication have been added in the revised version as follows;

"We found that the overall prediction skill is relatively low in warm season (June-September) with a larger spread compared with the cold season (October−May). This result is roughly consistent with the larger variance of the SIT anomaly in the warm season in the ESS (Fig. 5c). A large portion of the prediction skill at the lead times of 0−3 days can be explained by the persistency

effect based on the initial SIT (Fig. 6b). The contribution of the operational model on the forecast skill is less than 5% at shorter timescale (<3 days) (Fig. 6c), but the contribution of the operational model gradually increases at longer lead times except in May and October. In July, the contribution of the operational model on the prediction skill reaches ~15% at 7 day lead time. These results indicate that the operational model substantially improves the medium-range prediction skill of the SIT distribution in summer." (Line 314-324)

*3. Lines 134-138. Move the portion "In this . . .process [Startk et al. 2008]" into the first paragraph of section 2.*
**\*According to the reviewer's comment, we have moved these sentences into the first paragraph of section 2 (Lines 130-134).**

*4. Line 145. How is the 10-member ensemble produced?*
**\* To produce the ensemble members in the TOPAZ4 forecast system, the atmospheric forcing (e.g. wind speed), which is the ECMWF global atmospheric forecast data, as well as several parameter of sea ice model (such as e: the ratio of yield curve for rheology) are perturbed by adding stochastic forcing term due to inherent model errors [Evensen, 2003]. In this perturbation, the model error ($q_k$) is calculated based on the assumption that the perturbations of the forcing fields are related to red noise as follows;**

$$\vec{q}_k = \alpha \vec{q}_{k-1} + \sqrt{1-\alpha^2}\, \vec{w}_{k-1}. \tag{1.1}$$

**Where, $\alpha$ is lag 1 auto-correlation and $w_k$ is a sequence of white noise with the mean 0 and variance 1. This stochastic forcing term is added to the atmospheric forecast value and several parameters of sea ice model. In the revised version, we have added the essence of these descriptions as follows;**
**"To produce 10 ensemble members in the TOPAZ4 forecast system, the ECMWF global atmospheric forecast data as well as several parameters of sea ice model are perturbed by adding stochastic forcing term [Evensen, 2003]." (Lines 155-158)**

*5. Line 146. Spell out ECMWF.*
**\* According to the reviewer's comment, I have spelled out ECMWF in the first appearance of this manuscript as follows;**
**"…, forced at the surface by the European Centre for Medium-Range Weather Forecasts (ECMWF) operational atmospheric forecasts,…"(Line 96)**

**\* We apologize for the inappropriate number of forecast data of 259, which was not for 4 years (2013 to 2016), but for 5 years (2012 to 2016). On the other hand, we found that the prediction skills are strongly fluctuated before 2013 at initial step, after we have rechecked the PCC in each case (Fig. A1). According to coauthor's comment, such case is related to the free run without the initialization based on observational data (For example, 17th July 2014). Since these forecast data substantially reduce the initial prediction skill and increases its spread, we used the forecast data from 2014 to 2016 (Line 152) and removed the forecast data in July 2014 in the revised version. Consequently, the total of 150 cases was assembled during 4 years (2014-2016). I have modified the corresponding sentence in the revised version as follows;**

**"In this study, we excluded the forecast data in July 2014, because of a real-time forecast production incident (the forecast were in free-running mode then) [H. Engedahl, personal communication]. Since the forecast data were only provided weekly before 2016, the total of 150 cases was assembled during the study period." (Lines 159-162)**

**Based on this new forecast dataset, we recalculated the prediction skill of SIT, sea ice velocity, and wind speed by removing these spurious forecast data. Thus, Figures 6, 7, and 10 were changed in the revised version. Overall features of PCCs were not essentially changed, but the absolute values somewhat have been increased.**

[Figure]

**Figure A1. Daily means of the PCCs between forecast and analysis SIT at first step during 2013-2016.**

**\* According to the reviewer's comment, I have spelled out PIOMAS in this sentence as follows;**

**"…, we used the Pan-Arctic Ice Ocean Modeling and Assimilation System (PIOMAS) outputs,…" (Line 170-172)**

*8. Line 272. Change "completely" to "largely". The correlation shows that they are still related to some extent.*

**\* As the reviewer pointed out, our sentence is not accurate. According to the reviewer's suggestion, we have rephrased the corresponding sentence as follows;**

**"…, the predicted and analyzed sea ice velocities are largely unrelated." (Line 355)**

**\* I apologize for the typograph error. We have modified the figure number correctly as follows;**

**"…during the entire passage (Fig. 13a),…" (Line 455)**

**\* In this manuscript, in the case of the first word in sentences, we adopt the word "Figure". On the other hand, the word "Fig." is used in the case for the last word**

in sentences. The use of " Figure" and "Fig." appropriately depends on the rule of the manuscript format in this journal.

In addition to the revision based on the reviewer's comments, we also have revised the following items listed below;

1)   We have refined several sentences for clarification (e.g., Lines 146, 149, 186, 213).
2)   We removed the citation [Nakanowatari et al. 2017], which is it is a proceeding of Monbetu-2017 Symposium (Line 100) and the reference which is not cited in this paper [Nakanowatari et al. 2014] (Lines 667-669).
3)   We have updated the following reference information.

Yamagami A., Matsueda M., & Tanaka H. L. 2018. Predictability of the 2012 great Arctic cyclone on medium-range timescales, 15, 13-23, doi: 10.1016/j.polar.2018.01.002. (Lines 747-748)

---

## Author Comment (AC4) · 24 Apr 2018

*This manuscript investigates the forecast skill of the sea ice thickness distribution in the East Siberian Sea in early summer for a lead time from a few days to 10 days. The description and validation of the TOPAZ4 reanalysis utilized for this analysis are clear. They demonstrate the characteristic time evolution of the prediction skill and suggest the reasons for such as the abrupt reduction of the skill after 4 days. Their explanations by using simple models are reasonable and useful for the community of the Arctic sea ice monitoring and prediction. Therefore, I recommend this manuscript to be accepted for publication in the Cryosphere.*

*Please check the minor comments as described below.*
*L53: "CMIP" firstly appears here.*
**\* According to the reviewer's comment, we have corrected the corresponding part as follows;**
**", based on the Coupled Model Intercomparison Project Phase 5 (CMIP5)…" (Lines 55)**
**"…using the CMIP3 and CMIP5 global climate model simulations…" (Lines 561-562)**

*L212: The bias quantities described in this paragraph seem to depend on the definition of the ESS with negative biases in the north and positive biases in the south (Fig. 1c).*

*This should be mentioned here and the conclusion section. I believe it does not degrade the analysis of this study and help to avoid too much damaging the pedigree PIOMAS data.*

**\*As the reviewer pointed out, the definition of the ESS is crucial for our conclusion. To take care of this point, we have added the following sentence as follows;**
**"From the difference map of the climatological SIT between TOPAZ4 reanalysis data and PIOMAS output, the TOPAZ4 SIT is thicker near the coastal region with ∼50 cm (Fig. 1c), although the SIT in the offshore region is underestimated. These positive and negative biases are compensated each other and thus the mean bias of the TOPAZ4 SIT is 21 cm in July, which is smaller than those in winter (Table 3)." (Lines 242-247)**

**For the positive bias of the SIT in TOPAZ4 along the coastal region of the ESS, there is possibility that the SIT estimates (PIOMAS and CS2SMOS) used for the comparison are themselves underestimated. Schweiger et al. [2011] pointed out that the SIT of PIOMAS is underestimated by -17cm in the basin area of the Arctic Ocean including the Beaufort Sea where the heavy deformed sea ice formation occurs. Also, it was reported that the CS2SMOS data tend to show the underestimation in the region such where multi-year ice and first-year ice are formed, due to the spatio-temporal resolution of CryoSat-2 and SMOS and the merging algorithm [Ricker et al. 2017]. Since in the ESS, sea ice motion is strongly converged during winter [Kimura et al. 2013], there is possibility that the sea ice in the ESS is also heavily deformed to form the sea ice thicker than 1 m along the coastal region. In fact, our analysis based on the AIS data suggests that the SIT in excess of 100 cm is found in the coastal region of the ESS. Thus, for the precise evaluation of the SIT distribution in the ESS, the further improvement of ice-type as well as the accumulation of in-situ SIT measurement is needed. These discussions have been added in section 6 in the revised version (Lines 4853-496).**

*L230: I consider that the 2nd "errors" can be eliminated.*
**\* Through the revision, this sentence was removed in the revised version (Line 297).**

*L252: The abrupt reduction in October in Fig. 5 is not clear to me. Please chick this.*
**\* As the reviewer pointed out, the abrupt reduction of prediction skill in October is obscure in Fig. 5. We checked the prediction skill in October and found that it**

shows discontinuous change from the lead time of 3 to 4 days, but the reduction rate and the enhancement of the STD is smaller than those in May and July (Fig. A1). On the other hand, the prediction skill in September, in which the seasonal reduction rate of SIT is small, shows the abrupt reduction in Fig. 6. Thus, we removed the statement in the prediction skill in October in the revised version, and added some additional statements as follows;

"Such an abrupt reduction of the prediction skill and the enhanced standard deviation are also found in May and September, although the absolute values of the reduction rates are smaller than that in July. Since the influence of sea ice melt is small in these months (Fig. 5c), the abrupt reduction of early summer SIT prediction skill might be attributable to dynamical advection of sea ice." (Lines 328-332)

[Figure]

**Figure A1. PCCs between forecast and analysis SIT in TOPAZ4 in July (black), September (red), and October (blue) averaged on 2014-2016. Error bar indicates the standard deviation of the PCCs.**

In addition to this revision, we have merged Figures 6 and 7 in old version into Figure 7 in the revised version.

*L261: Please weaken the statement "the SIT distribution has a zonally homogeneous pattern".*
*According to the reviewer's comment, we have rephrased the corresponding sentence as follows;
"Since the SIT distribution has a tongue-like distribution (Fig. 5a),..." (Line 342)

*L272: "directed southward" should also be weakened if it is not exactly southward.*

**\*According to the reviewer's comment, the direction of predicted sea ice velocity is not exactly southward. To avoid the misleading, we have rephrased this sentence as follows;**

**"The resultant onshore anomaly of sea ice velocity leads to positive and negative anomalies..." (Line 355)**

*L277: "a deficiency at predicting Arctic cyclone". Please check this "at". (I am not a native English speaker and sorry if this is correct.)*

**\* As the reviewer pointed out, this preposition is not appropriate in this case. We have rephrased this as follows;**

**"… is related to a deficiency in the prediction of Arctic cyclone formation." (Line 360)**

*L312 and L313: I think the units "cm s-1" should be "m s-1". Please check them.*

**\* We apologize for the incorrect unit. We have corrected the unit by m s$^{-1}$ (Lines 396z-397)**

*L320: Since the authors describe on the reduction of the prediction skill in the 4th day, some words should be added to "remains at high level after the lead time of 4 days" on how high it is in order to avoid confusing.*

**\* As the reviewer pointed out, this description is misleading, because this sentence is somewhat inconsistent with the abrupt reduction of prediction skill at 4$^{th}$ day. Thus, we have appropriately revised the corresponding sentence as follows;**

**"It is interesting that the prediction skill of SIT in early summer remains ~0.9 at the lead times longer than 4 days (Fig. 7a), despite the poorer prediction skill… (Fig. 7b)." (Lines 404-406)**

**In addition to this revision, we have removed the related sentence in section 6 (Lines 525-527).**

*L358: "controlled by the weak skill of atmospheric prediction" is not clear to me.*

**\*As the reviewer pointed out, the prediction skill in winter is not necessarily controlled only by the atmospheric prediction skill but also ocean current change. Since the uncertainty of this discussion seems to be large, we have removed them in**

the revised version (Lines 439-445).

\* We apologize for the wrong Figure number. We have appropriately corrected the figure number in the revised version (Line 451).

\* We apologize for the wrong Figure number. We have appropriately corrected the figure number in the revised version (Line 455).

*L371: Please provide the significance of the difference between these two correlation coefficients if possible. Even if it is not significant, please do not consider to delete this interesting section. The sample number will be increased in the future to determine its significance as described in the final section. However, more careful discussion seems to be required for the conclusion in this section, since 1) the SIT and SIC time series can be resemble and 2) reproduction of SIC in the TOPAZ4 reanalysis is not validated in this study.*

\* As the reviewer pointed out, the difference between the correlations based on SIT and SIC is not so large (0.03, which accounts for only 5% of variance). In fact, the SIT is significantly correlated with SIC (r=0.86). We also admit that the number of sample is not enough to discuss the difference of the correlation relationship between SIT and SIC.

Our analysis based on the daily-mean AIS data might not be appropriate, which was noticed by my co-author during this revision process, because the vessel speed is fast to experience the multiple grids of TOPAZ4 SIT data during one day. To check the above possibility, we examined the statistical relationship between raw AIS data, whose time interval is about 2-3 hours, and daily mean SIT in TOPAZ4. The corresponding scatter plots of SIT and SIC to the corresponding vessel speed are shown in Figure A2, respectively. The correlation between the vessel speed and SIT is -0.56 (n=160), which is significant at 99% confidence level (Fig. A2a). On the other hand, the correlation between the vessel speed and SIC is -0.41 (n=160), which is insignificant at 99 confidence level. The scatter plots for SIC (Fig. A2b) indicates that the SIC value is somewhat insensitive to the vessel speed higher than 10 knot. Although the problem of sample size number still remains even in this analysis, these results support that the vessel speed was influenced by sea ice stress due to SIT and indirectly supports the reliability of the daily mean SIT of the

TOPAZ4 reanalysis data in the ESS in early summer. Thus, we have replaced Figure 12 in old version by Figure A2 and modified the corresponding sentence as follows;

"A joint statistical analysis of the daily mean SIT in the TOPAZ4 reanalysis and the vessel speed along the route indicates that vessel speed is significantly anticorrelated with SIT (–0.56) during the entire passage (Fig. 13a), significant at the 99 % confidence level based on a Monte Carlo technique [Kaplan and Glass, 1995]. We also examined the corresponding SIC data in TOPAZ4 reanalysis data, but the correlation between the vessel speed and SIC is -0.41 (Fig. 13b), which is insignificant at 99% confidence level. The scatter plots for SIC indicates that the SIC value is somewhat insensitive to the vessel speed higher than 10 knot. Although the problem of sample size number still remains in this analysis, these results support that the vessel speed was influenced by sea ice stress due to SIT and indirectly supports the reliability of the daily mean SIT of the TOPAZ4 reanalysis data in the ESS in early summer." (Line 453–462)

[Figure]

**Figure A2. Scatter plots of hourly vessel speeds (knots) and (a) daily mean SIT (cm) and (b) SIC (%) in TOPAZ4 reanalysis from 4–30 July 2014. In each panel, the regression line of vessel speed onto each variable is shown by broken line.**

In addition to the revision based on the reviewer's comments, we also have revised the following items listed below;

1) We have refined several sentences for clarification (e.g., Lines 146, 149, 186, 213).

2) We removed the citation [Nakanowatari et al. 2017], which is it is a proceeding of Monbetu-2017 Symposium (Line 100) and the reference which is not cited in this paper [Nakanowatari et al. 2014] (Lines 667-669).

3) We have updated the following reference information.

Yamagami A., Matsueda M., & Tanaka H. L. 2018. Predictability of the 2012 great Arctic cyclone on medium-range timescales, 15, 13-23, doi: 10.1016/j.polar.2018.01.002. (Lines 747-748)

---

## Author Response (AR2)

9 May 2018

**Dr. John Yackel**
**Editor, The Cryosphere**

Dear Dr. John Yackel

Thank you very much for your treatment of the article entitled "Medium-range predictability of early summer sea ice thickness distribution in the East Siberian Sea based on the TOPAZ4 ice-ocean data assimilation system" by Takuya Nakanowatari and co-authors (tc-2018-25). I understand that your final review of the revised manuscript is needed for the recommendation of publication in this journal.

After we submitted the revised manuscript in 24 April 2018, we found that the grant number shown in the acknowledgement is not correct. Therefore, we have modified the grant number in the revised version.

"…and JSPS KAKENHI Grant Numbers 17KK0014, 18H03745." (Line 517)

We also have refereed following reference which was recently published and highly related to our study as follows;

"Mohammadi-Aragh et al. [2018] suggest the dominant role of the chaotic behavior in atmospheric prediction skill on the short-term predictability of sea ice deformation in the Arctic Ocean." (Lines 84-86)

Mohammadi-Aragh M., Goessling H. F., Losch M., Hutter N., & Jung T. 2018. Predictability of Arctic sea ice on weather time scales. Sci. Rep., 8, 6514, doi:10.1038/s41598-018-24660-0.

The updated manuscript was uploaded through the portal page of "The Cryosphere".

We hope the manuscript has been improved satisfactorily for publication in The Cryosphere.

Sincerely yours,

Takuya Nakanowatari (nakanowatari.takuya@nipr.ac.jp)
Arctic Environment Research Center
National Institute of Polar Research
10-3, Midori-cho, Tachikawa-shi, Tokyo, 190-8518, Japan
Tel (81)-42-512-0763